# Monobody adapter for functional antibody display on nanoparticles for adaptable targeted delivery applications

C. Albert[1,10], L. Bracaglia[1,10], A. Koide[2,3], J. DiRito[4], T. Lysyy [4], L. Harkins[1], C. Edwards[4], O. Richfield[1,4], J. Grundler [1], K. Zhou[5], E. Denbaum[2], G. Ketavarapu [2], T. Hattori[2,6], S. Perincheri[7], J. Langford[4], A. Feizi [1], D. Haakinson[4], S. A. Hosgood[8], M. L. Nicholson[8], J. S. Pober[9], W. M. Saltzman [1], S. Koide [2,6] ✉ & G. T. Tietjen [1,4] ✉

Vascular endothelial cells (ECs) play a central role in the pathophysiology of many diseases. The use of targeted nanoparticles (NPs) to deliver therapeutics to ECs could dramatically improve efficacy by providing elevated and sustained intracellular drug levels. However, achieving sufficient levels of NP targeting in human settings remains elusive. Here, we overcome this barrier by engineering a monobody adapter that presents antibodies on the NP surface in a manner that fully preserves their antigen-binding function. This system improves targeting efficacy in cultured ECs under flow by >1000-fold over conventional antibody immobilization using amine coupling and enables robust delivery of NPs to the ECs of human kidneys undergoing ex vivo perfusion, a clinical setting used for organ transplant. Our monobody adapter also enables a simple plug-and-play capacity that facilitates the evaluation of a diverse array of targeted NPs. This technology has the potential to simplify and possibly accelerate both the development and clinical translation of EC-targeted nanomedicines.

Endothelial cells (ECs) are attractive therapeutic targets because they play an active role in a diverse array of diseases and are directly accessible to intravascular treatments[1,2]. Despite this potential, there are currently no EC-specific therapies in clinical use[3], a fact driven by the challenge of treating ECs without affecting other cell types. Endothelial-targeted nanoparticles (NPs) are a promising solution, as they have the potential to deliver a concentrated dose of various therapeutics (e.g. small molecule, nucleic acid, protein) directly to targeted ECs. In addition, NPs can protect the encapsulated therapeutic from degradation by the environment, facilitate its transport

across ECs membranes and sustain its release once inside the cells to prolong the duration of treatment[4]. Thus, effective targeting of NPs to the vascular endothelium within an organ of interest would create many new treatment possibilities across a variety of diseases.

The first hurdle to achieving EC-specific therapies is to localize the NPs to the tissue to be treated while avoiding losing NPs to the phagocytes of the liver and spleen[5,6]. In the field of organ transplant, we can circumvent this challenge by delivery in isolated organs during ex vivo normothermic machine perfusion (EVNMP). EVNMP is already in clinical use as a method to assess, preserve and potentially revive

[1]Department of Biomedical Engineering, Yale University, New Haven, CT, USA. [2]Perlmutter Cancer Center, New York University Langone Medical Center, New York, NY, USA. [3]Department of Medicine, New York University School of Medicine, New York, NY, USA. [4]Department of Surgery, Yale University, New Haven, CT, USA. [5]Department of Molecular Biophysics and Biochemistry, Yale University, New Haven, CT, USA. [6]Department of Biochemistry and Molecular Pharmacology, New York University School of Medicine, New York, NY, USA. [7]Department of Pathology, Yale University, New Haven, CT, USA. [8]Department of Surgery, University of Cambridge, Cambridge, UK. [9]Department of Immunobiology, Yale University, New Haven, CT, USA. [10]These authors contributed equally: C. Albert, L. Bracaglia. ✉e-mail: Shohei.Koide@nyulangone.org; gregory.tietjen@yale.edu

marginal organs with the goal of expanding access to organ transplantation[7]. The risk of dysfunctional inflammation is greater in marginal organs—organs from older and less healthy donors—and contributes to these organs being declined for transplant more frequently due to higher risk of post-transplant complications. Following organ recovery and preservation, ECs play a critical role in post-transplant pathologies associated with the dysfunctional inflammation. A single dose of therapeutics delivered in the forms of vascular-targeted NPs during EVNMP has the potential to reduce the immunogenicity of the graft and to provide several weeks of protection against dysfunctional inflammation during the post-transplant period when the organ is in its most vulnerable state[8]. Reduced endothelial activation has been demonstrated by treatment with anti-inflammatory molecules that inhibit NF-κB[9–11], mTOR[12], and complement[13–15] pathways as well as siRNA-mediated knockdown of molecules that activate the host's rejection response (e.g. MHC[8] and IL-15[16]). Thus, effective delivery of vascular-targeted NPs during EVNMP can both circumvent the challenge of organ-specific targeting and allow for ex vivo administration of therapies which could expand the pool of transplantable organs.

To target NPs at the cellular level, the most common strategy is to conjugate NPs with antibodies (Abs) or their derivatives that bind to an antigen on ECs. In a recent study, we demonstrated that PLA-PEG (poly[lactic acid]-poly[ethylene glycol]) NPs coupled with an Ab against CD31 bound nearly two orders of magnitude better than non-targeted NPs in cell culture, demonstrating the effectiveness of Ab-mediated NP targeting in a simple model[17]. However, when these NPs were delivered to transplant-declined human kidneys during EVNMP, the NP bound to only a minor fraction of the renal EC. This study illustrated the critical need to test NP technologies in complex, realistic settings[6], such as EVNMP of transplant-declined human organs. Moreover, our results demonstrated the need to substantially enhance the targeting efficiency of NPs in order to achieve sufficient therapeutic delivery necessary for translational relevance.

We posited that the lack of efficiency of our prior targeted NPs in human organ delivery was a result of the method by which the Abs were associated to the NP surface. Abs were conjugated to the surface of NPs through covalent coupling of the Ab free amines via the standard EDC-NHS chemistry[17]. This approach is widely used because of its simplicity and ease of use[18]. However, well-described limitations of this method include random orientation of the coupled Abs and potential modification of the lysine residues crucial for antigen binding. Both of these limitations lead to diminished binding potency due to a fraction of the Ab binding sites being unavailable[18–21]. To partially mitigate this problem, some have used Fc-binding proteins—such as protein A/G or FcRn to link the Ab and the NPs. These approaches only allow a partial control of the orientation as the proteins are themselves randomly oriented at the surface of the NPs via EDC-NHS chemistry[22–24]. This might negatively impact the surface density of the targeting Ab if not all adapters are available for binding.

As an alternative, targeting molecules can be engineered to support site-specific conjugation in a manner that avoids obscuring the sites of antigen binding (click chemistry, biotin-binding proteins, enzyme-based bioconjugation, and others). These alternative strategies are comprehensively reviewed by Sivaram et al.[18]. While effective, these approaches require modification of each Ab, which is labor intensive and costly. This added complexity and cost diminishes the ability to rapidly screen antibodies targeting different antigens. During the early development phase of a targeted NP, it is essential to identify the optimal antigen and targeting reagent for the given species, disease, organ and cell. Thus, a method to control Ab orientation without having to re-engineer each individual targeting molecule under consideration would be impactful. Here we describe an approach that combines simplicity, adaptability and efficiency by using unmodified antibodies as the targeting molecule.

In this work, we engineer a synthetic binding protein (i.e. a monobody or Mb) to act as a 'plug-and-play' adapter between NPs and targeting Abs. Our Mb adapter features a single cysteine positioned to enable thiol-based coupling to the NP surface in a controlled orientation. The Mb is further engineered to specifically bind the $F_C$ portion of IgG (Immunoglobulin G) Ab. In the resulting Ab-Mb-NPs, the Ab binding sites remain available to bind to the cells. In ECs cultured under flow, this strategy leads to a significant improvement in binding efficiency (>1000-fold) relative to standard EDC-NHS coupling of the same Ab. The potent targeting efficiency of these Ab-Mb-NPs also translates to delivery in transplant-declined human kidneys undergoing EVNMP. We further demonstrate the versatility of this approach by showing how the Mb adapter enables easy exchange of both the NP and targeting Ab without sacrificing binding efficiency or adding optimization steps. In other words, Ab-Mb-NPs are composed of three distinct blocks: (1) the drug encapsulating nanoparticle, (2) the Mb adapter, and (3) the targeting Ab. The Mb block serves as a universal adapter that links diverse combinations of NP and targeting Ab blocks that can be chosen to fit both the therapeutic need and corresponding target. The combination of potent targeting and versatility enabled by our Mb adapter has the potential to simplify and accelerate the preclinical development of vascular-targeted nanomedicines for diverse indications.

## Results

### Fc-binding monobody enables noncovalent coupling of cell-targeting antibody to NPs

In this proof-of-concept study, we developed a Mb that binds the Fc portion of mouse IgG1 (mIgG1) isotype as an antibody capture reagent (Fig. 1a). Mbs are the founding system of the many synthetic binding proteins based on a human fibronectin type III domain[25–27]. Mbs are stably folded, cysteine-free, β-sheet proteins developed from combinatorial phage-display libraries in which select residues are diversified using highly tailored amino acid compositions, followed by gene shuffling and further screening in the yeast-display format[28]. We developed a Mb, termed FCM101, that bound to mIgG1-Fc with high affinity (Supplementary Fig. 1a). We characterized its properties in the yeast display format where thousands of copies of a Mb are displayed on the yeast cell surface. This format provides a reasonable mimic of a Mb-conjugated NP. The interaction between FCM101 and mIgG1-Fc was stable and withstood competition with large excess of mIgG1, mIgG2a, rat IgG1, and bovine serum (Supplementary Fig. 1b). Surface plasmon resonance characterization of purified FCM101 revealed high-affinity binding, with very slow dissociation in the presence of other mouse IgG1 and only marginal binding to other IgG types tested (Supplementary Fig. 1c), further supporting the potency and selectivity of this Mb.

We next exploited the fact that Mbs are inherently free of cysteines to introduce a single cysteine residue in the extended C-terminal tail for site-specific conjugation (Fig. 1a) as has been previously shown in work by us and others[29–31]. We chose this location because it is at the opposite end of the Mb molecule from the diversified positions used for Fc binding (Fig. 1b). This design allows for site-specific Mb conjugation via a thiol-selective chemistry without impairing its Ab binding function (Fig. 1a). By conjugating FCM101 to the surface of dye-loaded PLA-PEG NPs through a terminal maleimide group on the PEG polymer, we have established a simple and robust method to conjugate Mb to NPs, which we term Mb-NPs hereafter. Ab-Mb-NPs were then produced in a second step by adding a mIgG1 Ab (either anti-CD31 or an isotype-matched control, creating CD31-Mb-NPs, or Iso-Mb-NPs) to a suspension of purified Mb-NPs (Fig. 1b). This two-step approach produced consistent association of both Mb and Ab with the NP surface (38 ± 9 μg of Mb/mg of NP; 47 ± 3 μg of Ab/mg of NP) as quantified by a pulldown assay (Fig. 1c, Supplementary Fig. 2, and Supplementary Fig. 3a). These amounts correspond to ~9 Mb

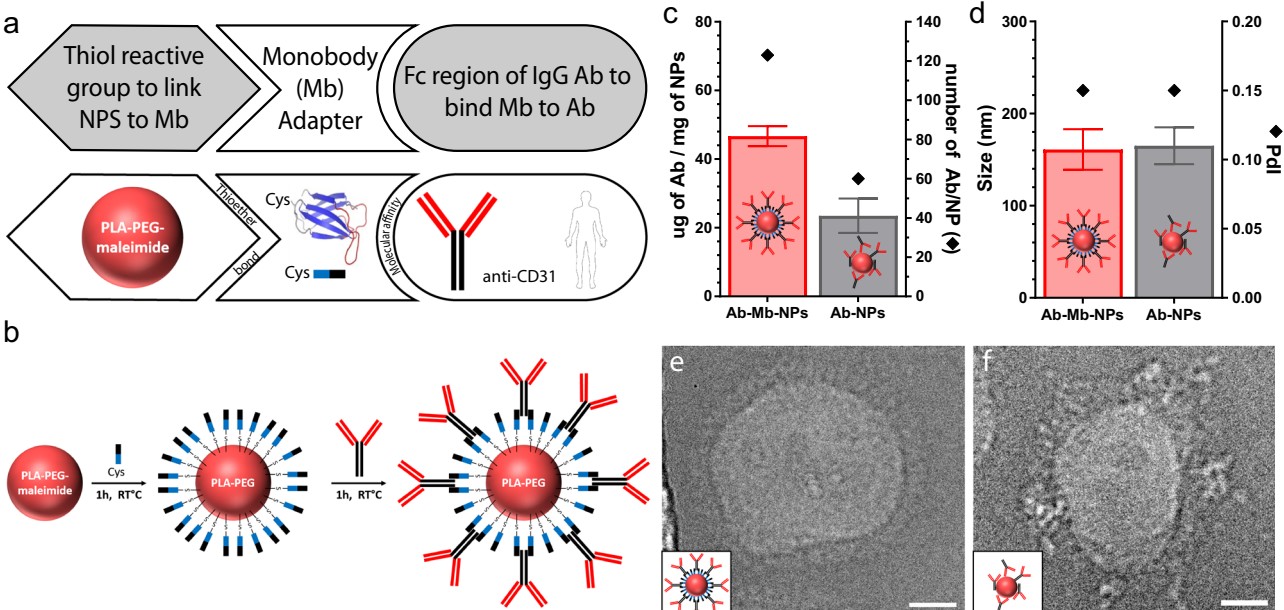

**Fig. 1 | Fc-targeting monobody couples IgG antibody to the surface of nanoparticles. a**, **b** Schematics illustrating the conjugation of an antibody (Ab) to the surface of a nanoparticle (NP) using a Fc-targeting monobody as an adapter. **c** Graph of the mass of Ab per mg of NP calculated from the concentration in the conjugation supernatant determined by SDS-PAGE electrophoresis and number of Ab per nanoparticle calculated with a NP of 100 nm in diameter and 1.25 g/cm³ in density (samples from $n > 10$ distinct NP preparations). **d** Graph of the size (hydrodynamic diameter) and PdI (polydispersity index) measured by dynamic light scattering (DLS) (samples from $n = 18$ distinct NP preparations). **e**, **f** Select Cryo-TEM images of Ab-Mb-NPs (**e**) and standard Ab-NPs (**f**) with clear protein visibility (not representative of sample average size). Cryo-TEM was performed once with 25 images taken per condition. Scale bar are 20 nm. Data are presented as mean values +/− SD. h hour, RT room temperature, PLA-PEG Poly (lactic acid)-poly (ethylene glycol).

molecules per 1 Ab molecule on the surface of the NPs (Supplementary Table 4). The hydrodynamic diameter and the polydispersity index (PdI) were conserved after each conjugation step demonstrating the stability of the NP formulation (Fig. 1d and Supplementary Fig. 3a). As expected, given the negative charge of the Mb with a pI of 5.54 estimated using the ProtParam tool (https://web.expasy.org/cgi-bin/protparam/protparam), a slight decrease in zeta potential from −5 ± 3 mV to −14 ± 5 mV was observed upon Mb conjugation (Supplementary Fig. 3a).

We next compared our Mb-mediated Ab capture method with direct, covalent coupling through free amines using the standard EDC/NHS chemistry (Supplementary Fig. 3b, c). Following our published method, we formulated PLA-PEG NPs conjugated to the same mIgG1 CD31 Ab[17]. These two types of NPs were similar in size, PdI, zeta potential, and dye loading (Fig. 1d and Supplementary Fig. 3a). However, the conjugation efficiency achieved with the Mb adapter approach was 47 ± 3 µg of Ab/mg of NP, substantially higher than the 23 ± 5 µg of Ab/mg of NP obtained with the covalent coupling method (Fig. 1c and Supplementary Fig. 3a). Cryo-TEM visualization showed a more ordered protein layer at the Mb-NP surface than the directly coupled NPs that exhibited a disordered and heterogenous layer (Fig.1e, f). No significant difference was noticed when the Ab-Mb-NPs were prepared as a small batch (few micrograms) or as a large batch (tens of milligrams) (Supplementary Fig. 5). Collectively, these data indicate that FCM101 enables robust capture of Abs to the surface of PLA-PEG NPs in an ordered fashion and with increased Ab density.

## Ab-Mb-NPs efficiently bind to endothelial cells in vitro and ex vivo

Ab-Mb-NP binding efficiency was assessed by targeting endothelial cells (ECs) with dye-loaded NPs in three experimental settings: static cell culture, microfluidic cell culture, and ex vivo perfusion of intact vessel segments. In each setting the binding efficiency was compared with dye-loaded Ab-NPs as a standard.

CD31-Mb-NPs and CD31-NPs both bound to human umbilical vein endothelial cells (HUVECs) under static culture conditions at higher levels than NPs coupled to a control antibody of the same isotype (Fig. 2a–d). However, CD31-Mb-NPs bound in significantly greater numbers, with mean fluorescence intensities (MFI) ~15 times higher compared to that of CD31-NPs when evaluated by flow cytometry (Fig. 2d). We next compared NP binding to HUVECs under physiological shear in a microfluidic chamber. Fluorescence microscopy (representative images in Fig. 2e, f) revealed a large increase in CD31-Mb-NPs bound to ECs compared with CD31-NPs. This result was confirmed by quantitative analysis which showed three orders of magnitude difference between CD31-Mb-NPs over CD31-NPs (Fig. 2g). Finally, NPs were administered during ex vivo perfusion of isolated human umbilical arteries, which preserves endothelial organization and 3D vascular morphology[32]. After 60 min of perfusion, substantially more CD31-Mb-NPs bound to ECs than CD31-NPs, as observed by confocal microscopy (Fig. 2h, i) and quantified by flow cytometry (Fig. 2j, k). In this system, the MFI of bound CD31-Mb-NPs is an order of magnitude higher than CD31-NPs. The difference in fold change between the three experiments presented here can likely be explained by differences in the experimental setting. For example, in cell culture under flow the NPs are constantly replenished as compared with a single administration in static cell culture where binding may be diffusion limited as the NPs are depleted in the un-stirred layer at the cellular interface. In the ex vivo vessel perfusion, the binding environment is much more complex than ones involving monolayers of cultured cells and introduces new challenges to binding that are not accounted for in cell culture systems. Regardless, our Ab immobilization method increased targeted delivery of NPs compared with the conventional direct coupling across all three experimental systems. We hypothesize that this improvement is due to substantial increases in the surface density of active Abs on the NPs (increased in number and by a more functional orientation), which has increased the instances of multivalent interactions between cell-surface antigen

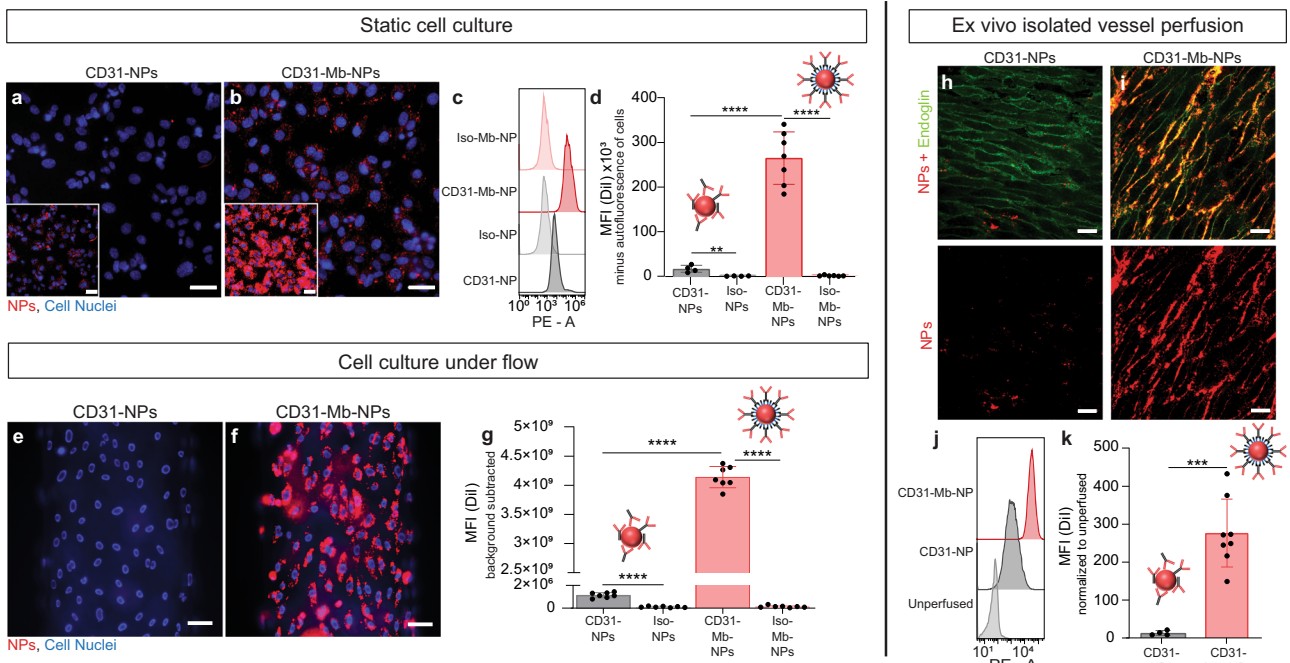

**Fig. 2 | The monobody adapter improves binding potency in cultured endothelium and isolated human vessels. a, b** Representative images of human umbilical vein endothelial cells (HUVECs) in static cell culture after incubation with either CD31-Mb-NPs (PLA-PEG-DiI NPs) or CD31-NPs; nanoparticles (NPs) are displayed in red. Scale bar is 50 μm. Inset images are copies of the larger image, displayed with a higher brightness to display CD31-NPs. **c** Representative histograms and **d** summary data of the mean fluorescence intensity (MFI) measured by flow cytometry of HUVECs after incubation. Each data point represents the averaged MFI from fluorescent NPs in three wells with 10,000 HUVECs captured/well (*n* = at least four indpendent experiments). **e, f** Representative images of HUVECs grown in microfluidic chambers after CD31-NPs (**e**) or CD31-Mb-NPs (**f**) were delivered under flow. Scale bar is 50 μm. **g** Summary data obtained from quantifying fluorescence signal from NPs bound to HUVECs in the microfluidic chamber using fluorescence microscopy. Each data point in **g** represents an image (*n* = at least six images). **h, i** Representative *en face* images of human umbilical arteries perfused with CD31-NPs (**h**) or CD31-Mb-NPs (**i**). Anti-CD105-FITC is shown in green, NPs are shown in red. Scale bar is 20 μm. **j** Representative histograms show relative levels of NP fluorescence signal on cells isolated from perfused vessels. **k** Summary of MFI measured on cells isolated from vessels perfused with CD31-NPs or CD31-Mb-NPs (*n* = at least four independent experiments). Statistical significance is shown between groups with ****$p < 0.001$ and ***$p < 0.001$ was found using multiple two-tailed *T*-tests with a Bonferroni correction. Data are presented as mean values +/− SD. Source data are provided as a Source Data file. Mb monobody.

and NPs. Increased multivalency is expected to increase binding efficiency exponentially[33,34].

## The Mb adapter allows versatility in NP design and function

We envisioned that another strength of the Mb adapter is the modularity it provides to the formulation, both for different targeting Abs and types of NPs. We first tested this concept by targeting the same Mb-NP to different target molecules by coupling either a CD31 Ab or an ICAM2 Ab and compared the binding efficiency in human umbilical arteries using an ex vivo isolated vessel perfusion system[32]. The ICAM2-Mb-NPs efficiently bound to ECs, although their level of binding was 2.5 times lower than CD31-Mb-NPs (Fig. 3a–e), a ratio consistent with the different levels of these antigens in umbilical arteries. We then assessed the adaptability to ECs between different species by coupling Mb-NPs to mIgG1 antibodies against porcine CD31. Mb-NPs conjugated to either anti-porcine CD31 or anti-human CD31 showed consistent targeting efficiency only when NPs were added to cells of their respective targeted species (Fig. 3f–k). Together, these experiments demonstrate the simplicity of Ab substitution with the Mb adapter, enabling systematic investigation to optimize the selection of antigen and/or antibody for a specific tissue or application.

The Mb adapter can potentially be attached to a variety of NPs comprised of different materials and encompassing different therapeutic cargo, provided that a thiol reactive group is available on the NP surface. Many polymeric NPs are prepared by emulsion-evaporation, which typically requires a surfactant to stabilize the NP surface. In that case, any reactive groups of the polymer are masked by the surfactant, thereby preventing conjugation. One such polymer system

we have previously used for efficient delivery of nucleic acids[8,35,36] is poly(amine-co-ester) [PACE]. To adapt the Mb coupling system to PACE NPs, we grafted a thiol reactive group (here, vinyl sulfone (VS)) directly onto a commonly used surfactant, poly(vinyl alcohol) (PVA), producing poly(vinyl alcohol-vinyl sulfone) (PVA-VS) (Fig. 4a). Dye-loaded PACE-NPs stabilized with PVA-VS were conjugated to Mb and then coupled with Ab. The resulting NPs showed no significant change to their hydrodynamic diameter (≈340 nm) and PdI (≈0.17) as a result of coupling (Supplementary Table 6). An expected decrease of zeta potential was observed from ~ +20 mV to ~ −20 mV after the Mb and Ab were attached to the NPs. After 2 h of incubation on HUVECs, the amount of Ab-Mb-PACE-NPs bound to cells was significantly higher when coupled with a CD31 Ab as compared to a control, non-targeting Ab (Fig. 4b–e). PVA-VS can be used with other polymers formulated with emulsion-evaporation (such as PLGA for example). Thus, this method for indirect coupling of the Mb to NPs via PVA-VS substantially expands the applicability of the Mb adaptor to a diverse array of NPs.

## Comprehensive vascular coverage achieved with Ab-Mb-NPs during ex vivo perfusion of human kidneys

Our prior experience suggested that selective targeting to ECs in culture was much easier to attain than in human kidneys subjected to EVNMP[17]. To assess the clinical applicability of Mb coupling technology, we next tested the binding efficiency of Ab-Mb-NPs in human kidneys during EVNMP. EVNMP has emerged as an alternative to static cold-storage of organs procured for transplantation[37–40]. In this procedure, normothermic and normoxic conditions are reintroduced to the organ, prolonging the assessment period and providing privileged

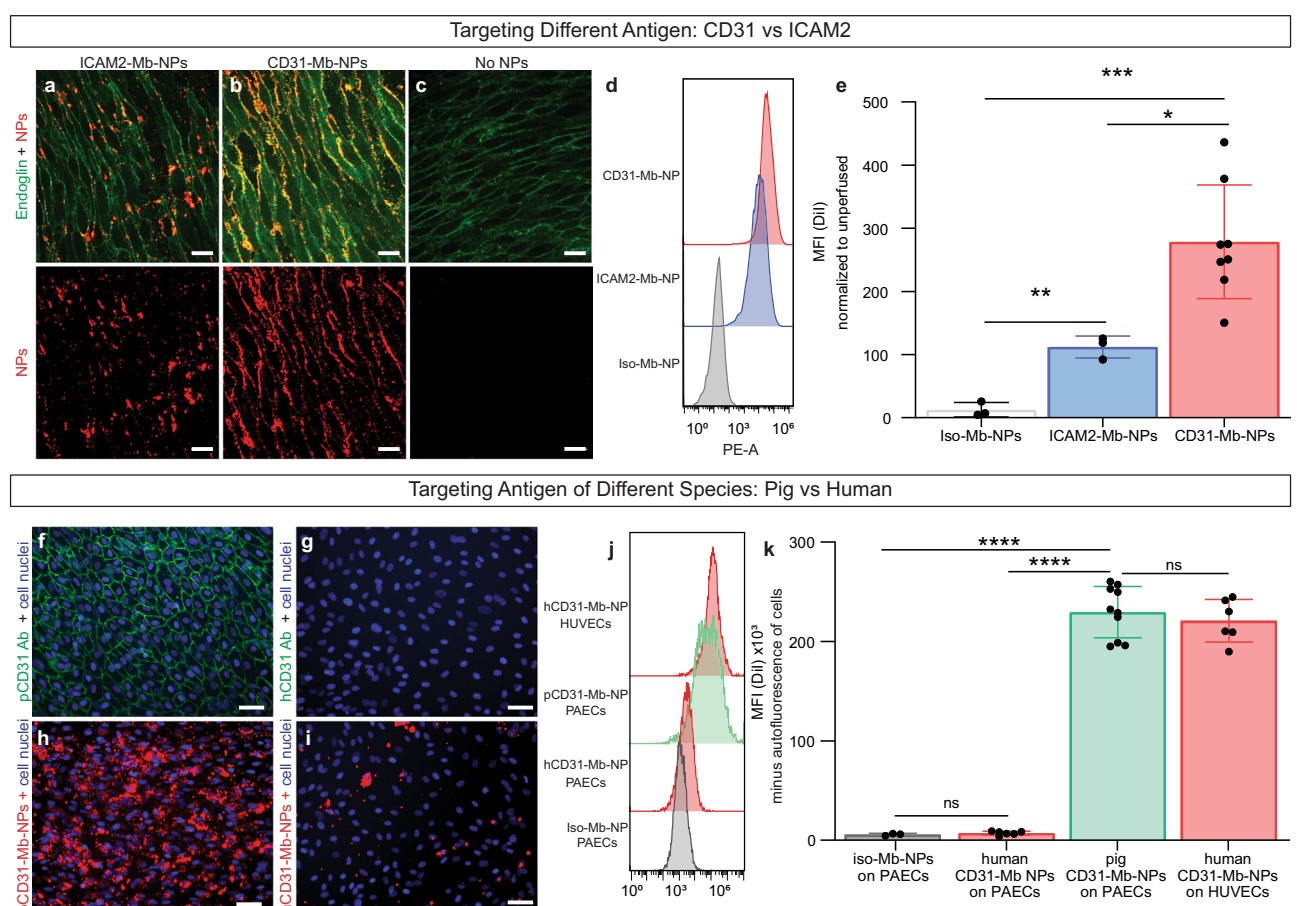

**Fig. 3 | The monobody adapter enables adaptability for antibodies targeting different human proteins or endothelial cells from different species.**
**a–c** Representative *en face* confocal images of a section from an isolated umbilical artery perfused ex vivo with ICAM2-Mb-NPs (**a**), with CD31-Mb-NPs (**b**) and without NPs (**c**). Tissues were stained with anti-CD105-FITC, depicted in green. NPs are depicted in red. Each image is displayed with a color merged view and accompanied with a red only image below. Scale bars are 20 μm. **d** Representative flow cytometry histograms and **e** summary data of mean fluorescence intensity (MFI) of fluorescent NPs bound to endothelial cells isolated from the umbilical artery after ex vivo perfusion (error bars represent SD). Each data point represents the MFI for a single vessel (*n* = at least three independent experiments). CD31-Mb-NPs data are the same as displayed in Fig. 2 K. **f–i** Representative fluorescent images of pig aortic endothelial cells (PAECs) incubated with either fluorescent anti-pig-CD31 antibody

(Ab) (**f**), fluorescent anti-human-CD31 Ab (**g**), anti-pig-CD31-Ab-Mb-NP (**h**) or anti-human-CD31-Ab-Mb-NP (**i**). Fluorescent antibodies and NP are respectively depicted in green and in red. Scale bars are 50 μm. **j** Representative flow cytometry histograms and **k** summary data of the averaged MFI of fluorescent NP bound to PAECs or human umbilical vein endothelial cells (HUVECs) in vitro (error bars represent SD). Each data point represents the averaged MFI from three wells with 10,000 HUVECs captured/well (*n* = at least three independent experiments). Statistical significance is shown between groups with ****$p < 0.0001$, ***$p < 0.001$, **$p < 0.01$, *$p < 0.05$ and ns where no significant difference was found using multiple two-tailed *T*-tests with a Bonferroni correction. Data are presented as mean values +/− SD. Source data are provided as a Source Data file. Mb monobody, NP nanoparticle, pCD31 anti-pig-CD31 Ab, hCD31 anti-human-CD31 Ab, iso isotype.

access to the organ for treatment prior to transplantation. NPs were introduced by bolus injection in kidneys undergoing EVNMP and were allowed to circulate for a period of 4 h. In this study, we enrolled 9 human kidneys which were declined for transplant and procured for research. Donor demographic information is provided in Supplementary Table 7 and histopathologic assessment of each organ is provided in Supplementary Fig. 8.

Targeted Ab-Mb-PEG-PLA-NPs (with either CD31 Ab or isotype) and untargeted PLA-PEG-NPs, each loaded with a different dye, were introduced together into the perfusate of each kidney. In an effort to exclude nonspecific NP accumulation in red blood cell aggregates—which have been previously found in kidneys from this marginal subset[17,41]—areas where targeted (red) and untargeted (green) NPs co-localized at equal levels were removed from analysis (Supplementary Figs. 9 and 12). Areas of specific binding from either CD31-Mb-NPs or Iso-Mb-NPs were then quantified (Fig. 5, Supplementary Figs. 9, 10). Comprehensive and abundant binding of CD31-Mb-NPs was observed in both large vessels (Supplementary Fig. 11). and in the microvasculature (Fig. 5a, c, f). Calculating the area of vascular staining that

co-localized with targeted NPs suggests that 49% of the available microvascular area outside the glomeruli is covered with CD31-Mb-NPs, compared to 1% coverage with Iso-Mb-NPs (Fig. 5e). This coverage is enhanced in glomeruli regions, with 79% of the available area covered with CD31-Mb-NPs and 10% with Iso-Mb-NPs (Fig. 5h). The increase in covered area by CD31-Mb-NPs between the microvasculature and the glomeruli could be explained by (1) a local increase in NP concentration in the glomeruli as the blood is filtered and concentrated, (2) a depletion of NPs from the perfusate as blood flows from glomeruli to the rest of the microvasculature, and (3) the smaller size and tortuosity of the glomerular microvessels could increase the probability of a binding event.

In our post perfusion analysis, Kidney 2—which had the lowest binding—was discovered to have extensive glomerular sclerosis (>10% determined with H&E staining, Supplementary Fig. 8), as well as interstitial fibrosis and other injury. This may indicate poor vascular patency throughout the kidney which may have diminished circulation. In this event, the exposure of the NPs to ECs might have been reduced along with the achievable coverage. If this kidney is

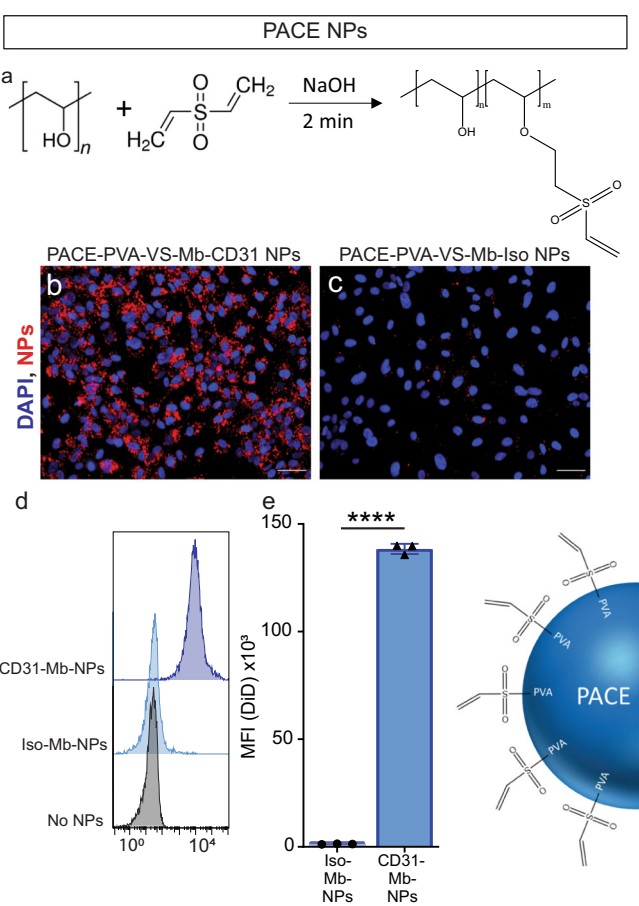

**Fig. 4 | The monobody adapter is compatible with nanoparticle systems prepared by emulsion-evaporation using poly(vinyl alcohol-vinyl sulfone).**
**a** Schematic of the reaction to prepare poly(vinyl sulfone). Fluorescent images of human umbilical vein endothelial cells (HUVECs) treated in vitro with PACE-PVA-VS-Mb-CD31 NPs (**b**) or PACE-PVA-VS-Iso NPs (**c**). Scale bars, 50 μm. The cell nuclei were stained with DAPI, depicted in blue. The NPs are depicted in red.
**d** Representative flow cytometry histograms and **e** mean fluorescence intensity (MFI) values of fluorescent Ab-Mb-PACE-NP bound to HUVECs in vitro (error bars represent SD). Each data point represents the averaged from three wells with 10,000 HUVECs captured/well (n = at least three independent experiments). Statistical significance between treatments is denoted with ****$p < 0.001$, by a multiple two-tailed $T$-tests with a Bonferroni correction. Data are presented as mean values +/− SD. Source data are provided as a Source Data file. PACE Poly(amine-co-ester), PVA Poly(vinyl alcohol), VS vinyl sulfone, NP nanoparticle, Ab antibody, iso isotype Ab, Mb monobody.

removed from analysis, the average coverage of CD31-Mb-NPs is increased to 56% in the microvasculature outside the glomeruli and 92% in the glomeruli. Considering the full data set of the six kidneys, the area of coverage achieved is much improved from the level we were able to obtain in previous reports using Ab-NPs produced by a more conventional covalent coupling method[17]. By re-analyzing the images taken in this previous study using the current analytical method, we estimate that the area of coverage in the glomeruli was ~31% using standard CD31-NPs, compared to ~79% using CD31-Mb-NPs. The difference is even more striking in the vasculature outside the glomeruli where coverage increased from 1.2% using the CD31-NPs to 49% using CD31-Mb-NPs (Fig. 5 and Supplementary Fig. 15). This improvement was confirmed experimentally by perfusing an additional pair of kidneys from the same donor with either Mb-conjugated or EDC-NHS-conjugated CD31 Ab to NPs, as well as in a single kidney using two distinct NP fluorescent dyes (Supplementary Figs. 13 and 14).

## Discussion

Here we have described an approach to couple Abs to NPs using an engineered Mb as an adapter (Fig. 6). The resulting Ab-Mb-NPs address the key challenges facing targeting NPs in that they are simple to formulate, adaptable, and highly efficient in binding ECs even in complex settings. The improvement in binding (>1000-fold in cell culture under flow) of the Ab-Mb-NPs over standard Ab-NPs prepared by EDC-NHS chemistry can likely be explained by (1) an increased number of Abs and (2) a controlled orientation of those Abs on the surface of the Ab-Mb-NPs. Ab-Mb-NPs can also target different cell surface molecules by simply swapping any available Ab of the same isotype or combining multiple of these Abs to hit different targets at once. This demonstrates an improvement over the strategies that require direct Ab modification. The exchange of the targeting Ab could be used to facilitate the screening and discovery of the optimal antigen to target for a particular tissue, or could be motivated by a transition between pre-clinical models (e.g. porcine to human organs). In addition to the versatility of the targeting molecule, our design of the FCM101 construct with a single Cys residue enables its conjugation to a variety of NPs presenting a thiol reactive group on their surface. This approach grants the ability to rapidly screen for the best vehicle for a particular therapeutic without further modifications to the targeting strategy and expands the possible applications. In the end, this approach can be thought of as a building-block design where each critical component—therapeutic agent, nanoparticle and antibody—can be switched without modifying the other blocks. Thus, we envision that our approach will facilitate more comprehensive screening for the optimal combination of these three components.

In the work shown here, we have utilized FCM101 which is specific for the mouse IgG1 isotype. Mouse IgG1 antibodies against human antigens are readily available, making this an ideal design to develop and demonstrate the approach. To minimize immune-reactions in humans, we envision the use of human or humanized Abs and a Mb adapter specific to human IgG Fc. Just as human Abs can be developed using many modern technologies, Mbs to human Fc can be developed following the well-established pipeline[26,28]. Furthermore, we envision that the potential problem of Ab displacement with endogenous Abs can be eliminated with a monobody adapter specific to a mutant Fc of human IgG (e.g. the so-called LALA variant that is commonly used for therapeutic antibodies[42]). We expect Mb adapters to minimally contribute to the overall immunogenicity of NPs based on clinical trials with a related molecule (PEGylated Adnectin) (NCT02515669, NCT03984812)[43,44].

Ab-Mb-NPs also resulted in excellent vascular area coverage when perfused in human kidneys during EVNMP. This result is encouraging for future applications of therapeutic delivery to marginal organs, such as the kidneys enrolled in this study. Targeted NPs to treat marginal organs prior to transplant could reduce the risk of dysfunctional endothelial cell inflammation and associated recruitment of immune cells, rendering those organs safer to transplant and thereby increasing the number of transplantable organs. Management of peri-operative graft injuries, e.g., due to ischemia/reperfusion or to binding of graft-reactive alloantibodies in pre-sensitized recipients, with therapeutic NPs likely will require that a critical number of ECs are exposed to enough NPs to reach an effective therapeutic dose. In the context of EVNMP, this is only possible if NP binding efficiency is strong in complex 3D environments involving flow. In fact, developing NPs in this setting will not only enable clinical translation in the context of ex vivo repair of marginal organs for transplant, but can also provide new insights relevant to intra-arterial or systemic delivery in the context of native kidney disease. Recent advances in EVNMP make it possible to sustain human organs outside the body for as long as 1 week[45], which may allow us to evaluate safety and efficacy of vascular-targeted

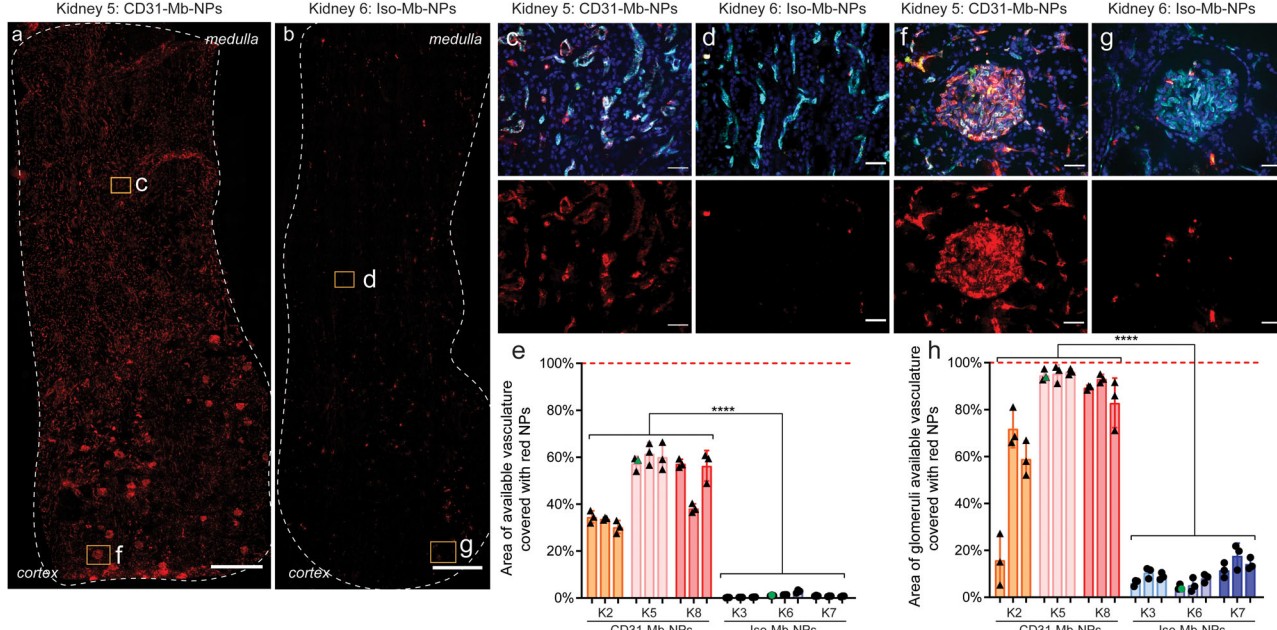

**Fig. 5 | CD31-Mb-NPs specifically and efficiently accumulate throughout human renal endothelium during ex vivo organ perfusion. a**, **b** Representative kidney biopsy slices (10 μm thickness) taken from organs after a 4 h ex vivo perfusion with either CD31-Mb-NPs (**a**) or Isotype (iso)-Mb-NPs (**b**). Images show nanoparticles (NPs) in red with high levels of accumulation throughout the biopsy sample when NPs are targeted using CD31 (**a**). The biopsy perimeter is outlined in white, and the cortex- and medulla-facing-ends are labeled for orientation. Yellow boxes outline the location of the high magnification images defined in the rest of the figure and are labeled with the corresponding letter. Scale bar is 1 mm. **c**, **d**, **f**, **g** Representative high magnification images of microvasculature and glomeruli display NPs in red, vascular cells in cyan, cell nuclei in dark blue, and untargeted NPs in green. Each image is displayed with a color-merged view and accompanied with a red-only image below. Scale bar is 50 μm. Kidneys perfused with CD31-Mb-NPs have much higher NP coverage (61% in C and 92% in glomeruli of **f**) compared with kidneys perfused with isotype-Mb-NPs (1.8% in **d** and 3% in glomeruli of **g**). **e**, **h** Summary quantification of the red NP coverage on available microvascular structures (**e**) and glomeruli (**h**) with $n = 3$ tissue slices. Three biopsies were taken from each of six kidneys after the 4 h ex vivo perfusion, and are grouped together by the bars for representation. Three slices of 10 μm thickness, separated by 100 μm, were sampled from each biopsy. Each data point represents the area of available vasculature determined to have NP presence by fluorescence signal in a slice (~45 mm² area, or 300, 20× images) (**e**) or ~24 glomeruli (**h**), divided by the total area of available vasculature. Data points in green represent the data collected from the tissue slices shown in **a** and **b**. Statistical significance between treatments is denoted with ****$p < 0.001$, by an unpaired two-tailed T-test. Data are presented as mean values +/− SD. Source data are provided as a Source Data file. Mb monobody.

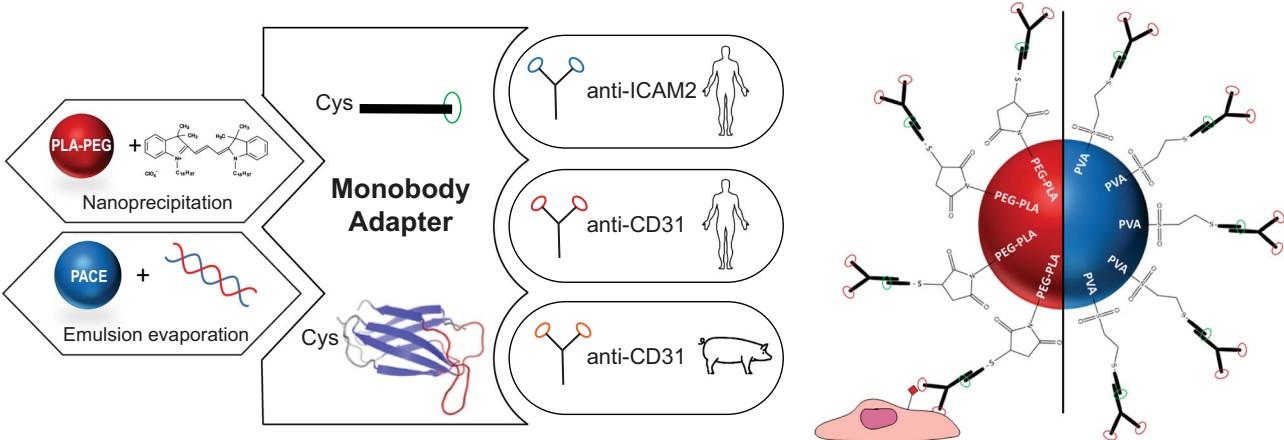

**Fig. 6 | Summary schematic of targeting strategy using the monobody adapter.** PACE Poly(amine-co-ester), PVA Poly(vinyl alcohol), VS vinyl sulfone.

nanomedicines in a human context prior to exposing patients to risk in human clinical trials.

Ab-Mb-NPs have shown strong cellular binding and possess modularity to support rapidly developing applications and their progression through experimental platforms. The Ab-Mb-NPs have the potential for almost immediate application in organ transplantation, and we are optimistic that this technology can be further developed for systemic delivery in a wide array of disease indications.

## Methods

### Chemicals

Poly (lactic acid)-poly (ethylene glycol) (PLA-PEG) block copolymer (Mw 16:5 kDa) were purchased from PolySciTech with different end group on the PEG block: carboxylic acid (PLA-PEG-COOH, AI030), methoxy (PLA-PEG-m, AK054) or maleimide (PLA-PEG-mal, AI065). Divinyl Sulfone (DVS, at 1.177 g/mL, <500 ppm hydroquinone as inhibitor), poly(vinyl alcohol) polymer (PVA 31–50 kDa, 87–89%

hydrolyzed), sodium hydroxide (NaOH, reagent grade >98% anhydrous), chloridric acid (HCl, 37%), pentadecanolide (98%), N-Methyldiethanolamine (>99%), diethyl sebacate (98%), lipase acrylic resin from *Candida antartica* (>5,000 U/g, recombinant, expressed in *Aspergillus niger*), gold nanoparticles maleimide functionalized (30 nm) and dexamethasone were acquired by Sigma-Aldrich. Deuterium oxide ($D_2O$, for NMR, 99,8 atom%D), phenyl ether (99%) were obtained from ACROS organic. 1,1'-dioctadecyl-3,3,3',3'-tetramethylindocarbocyanine perchlorate (DiI), Hoechst 33258, DL-dithiolthreitol (DTT), LDS Sample buffer (4x), NuPAGE™ SDS Running Buffer (20×), Fisherbrand superfrost microscope slides, cover slips (22 × 40 mm) were purchased from Thermo-Fisher Scientific. 1,1'-dioctadecyl-3,3,3',3'-tetramethylindodicarbocyanine, 4-chlorobenzenesulfonate salt (DiD) was obtained from Botium. Dulbecco's phosphate buffered saline (DPBS), medium 199 (M199), fetal bovine serum (FBS), penicillin-streptomycin, L-glutamine and TryPLE were acquired from Gibco. Coomassie Brilliant Blue R250 was obtained from OmniPur. Purified anti-human-CD31 monoclonal antibody (PECAM-1, IgG1 mouse, clone WM59), purified mouse IgG1 K isotype control antibody (clone MOPC-21) were purchased from Biolegend. Purified and FITC labeled anti-porcin-CD31 monoclonal antibody (PECAM-1, IgG1 mouse, clone LCI-4), purified anti-human-CD102 monoclonal antibody (ICAM-2, IgG1 mouse, clone BT-1), Alexa Fluor 488 labeled anti-human-CD105 monoclonal antibody (endoglin, IgG1 mouse, clone SN6) FITC labeled anti-human-CD31 monoclonal antibody (PECAM-1, IgG1 mouse, clone WM59), Alexa Fluor 647 labeled rabbit anti-Mouse IgG (H + L) cross-adsorbed secondary antibody were acquired from Invitrogen. BUV395 labeled anti-human-CD105 monoclonal antibody (endoglin, IgG1 mouse, clone 266) was obtained from BD Bioscience. Acetic acid glacial was purchase from RICCA. Methanol and dimethyl sulfoxide (DMSO) were acquired from JT Baker. Ethanol was obtained from AmericanBio. Human Umbilical Vein Endothelial Cells (HUVECs) were obtained from the Yale Vascular Biology and Transplantation tissue culture core laboratory, where they were isolated from fresh umbilical veins by treatment with collagenase. Porcine Aortic Endothelial Cells (PAECs) were purchased from Cell Applications, Inc. Endothelial Growth Supplement (ECGS) was acquired from Corning. Porcine EC Basal Medium with Porcine EC Growth Supplement was obtained from Cell Applications. Collagenase II was purchased from Worthington Biochemical. Human fibronectin was acquired from R&D. DyLight 649 labeled *Ulex europaeus* type I lectin (UEA I), Fluorescein labeled UEA1, antifade mounting medium with 2-(4-Amidinophenyl)-6-indolecarbamidine dihydrochloride (DAPI) were obtained from Vector Laboratories. Ringer's lactate (LR) solution and sodium bicarbonate (8.4%) were purchased from Hospira. Mannitol (10%), Glucose (5%) and Ringer's solution were acquired from Baxter Healthcare. Heparin (1000 IU/ml) was obtained from McKesson Medical-Surgical.

## Monobody development

Mouse IgG1 Fc protein with C-terminal Avi-tag and His6 tag was produced in ExpiCHO cells (ThermoFisher) and purified using Ni-affinity chromatography. The mouse Fc protein was enzymatically biotinylated using purified BirA enzyme. This protein was used to screen monobody phage-display libraries[28,46]. After four rounds of phage selection, the sorted pools were subcloned into a yeast display vector after recombination of 5′ and 3′ fragments, followed by two rounds of sorting in the yeast display format[28]. The monobody clone, FCM101, was validated for target binding using yeast display[28,46]. FCM101 with an N-terminal tag containing His6, FLAG epitope, a TEV cleavage site and a C-terminal tag containing a cystein residue were expressed using the pHFT vector in *Escherichia coli* and purified using nickel-affinity chromatography[28,46]. The amino acid sequence of FCM101 is shown below with the N- and C-terminal tags underlined.

MKHHHHHHSSDYKDDDDKGENLYFQGSVSSVPTKLEVVAATPTSLL ISWDAPAVTVYYYVITYGETGGNSPVQEFTVPGSKSTATISGLKPGVDYTI TVYAGYGSGGYYSPISINYRTEIDKC. The expression vector for FCM101 has been deposited to Addgene.

## PACE production

Poly(amine-co-ester) (PACE) was synthesized following previously published protocols by Kauffman and Piotrowski-Daspit[36]. Briefly, monomers (1.180 g MDEA, 2.558 g DES and 3.570 g PDL) together with enzyme (0.731 g lipase) were massed to obtain a molar ratio of 3:2:1. These materials were combined in a round bottom flask together with 14.6 g phenyl ether. Oligomerization was carried out under argon for 20 h. Polymerization was then carried out under vacuum for 48 h. The polymer was purified with three hexane washes to remove phenyl ether, and dissolved in DCM to remove the enzyme beads. Finally, DCM was evaporated under vacuum pressure using a rotary evaporator. The final product was characterized using NMR.

## PVA-VS synthesis

PVA-VS polymer was prepared as described by Raudszus et al.[47]. The excess of divinyl sulfone was removed by dialysis for 24 h against water (Slide-A-Lyzer®Dialysis Cassette, 10 000 MCWO, Thermo Scientific). The resulting PVA-VS was lyophilized and characterized by 1H NMR in D2O (Agilent DD2 400 MHz).

## Nanoparticles preparation and characterization

PLA-PEG NPs were prepared following a nanoprecipitation method[17]. PLA-PEG-COOH NPs were prepared with PLA-PEG carboxylic acid terminated polymer and PLA-PEG-mal NPs were prepared with a blend of methoxy and malemide terminated PLA-PEG polymer at a 1:1 weight ratio. Briefly, the polymer was dissolved in DMSO at 50 mg/mL and mixed with lipophilic fluorescent dye (DiI) at a dye:polymer weight ratio of 0.5%. This organic phase was then added dropwise with a glass pasteur pipette into distilled water under vigorous agitation (700 rpm, organic:aqueous phase volume ratio of 1:4). The organic solvent was subsequently removed using a 50 mL amicon filter tube (Amicon® Ultra-15 Centrifugal Filter Unit, MCWO 10000, Sigma Aldrich) for small batches or using a tangential flow filtration system (Spectrum® KrosFlo® KR2i TFF System, Repligen) equipped with a Microkros column (Microkros 20 cm 500 K MPES 0.5 mm MLL X FLL, C02-E500-05-N, Repligen) for big batches (more than 50 mg). The NPs suspension in DI water was flash frozen in liquid nitrogen and stored at −80 °C before use.

PACE-PVA-VS NPs were prepared following an emulsion-evaporation method[36]. The polymer was dissolved in DCM at 50 mg/mL and mixed with lipophilic fluorescent dye (DiD) at a dye:polymer weight ratio of 0.5%. This organic phase was then added dropwise with a glass pasteur pipette into a PVA-VS solution at 5% (w/w) under vigorous vortexing (organic:aqueous phase volume ratio of 1:2). The resulting emulsion was then transferred into a beaker containing 3 times its volume of PVA solution at 0.3% (w/w) under agitation. After 1 min the organic solvent was evaporated using a rotary evaporator. The excess of PVA and PVA-VS was washed by two centrifugations (18,000 × g, 45 min, 4 °C). The pellet was redispersed in DI water, flash frozen in liquid nitrogen and stored at −80 °C.

The size and potential zeta of the NPs were measured by dynamic light scattering (Zetasizer Nano ZS, He-Ne laser 633 nm, 173°, Smoluchowski equation, Malvern Panalytical).

## Ab conjugation to nanoparticles using EDC-NHS chemistry

PLA-PEG-COOH NPs were conjugated to targeting antibodies (IgG1 mouse anti-hum-CD31 Ab or IgG1 mouse Isotype Ab) as previously described[17] using 1-ethyl-3-(3-dimethylaminopropyl) carbodiimide hydrochloride (EDC) and N-hydroxysuccinimide (NHS) chemistry

resulting in Ab-NP (CD31-PEG-PLA-NP and Iso-PEG-PLA-NP). NPs are flash frozen in liquid nitrogen and stored at −80C until use.

## Ab attachment to nanoparticles using Mb-adapter technology

In a first step, PLA-PEG-mal NPs and PACE-PVA-VS NPs were conjugated with the Mb using the thiol reaction between the thiol reactive groups on the surface of the NPs (maleimide or vinyl sulfone) and the single terminal cysteine of the Mb. The reaction happened at room temperature with a mild mixing (Orbit Shaker, 250 rpm) for 1 h with a NPs:Mb weight ratio of 10:1. The excess of Mb was removed by centrifugation (15 min, $21,000 \times g$, 17 °C) and the NPs redispersed in PBS at a concentration of 5 mg/mL. In a second step, the Ab were attached to the surface of the NPs using the potent and strong binding affinity of the Mb for the Fc region of the IgG1 mouse of the Ab. The reaction happened at room temperature with a mild mixing (Orbit Shaker, 250 rpm) for 1 h with a NPs:Ab weight ratio of 18:1. The excess of Ab was removed by centrifugation (15 min, $21,000 \times g$, 17 °C) and the NPs redispersed in PBS at a concentration of 5 mg/mL. The resulting NP were named Ab-Mb-NP (Ab-Mb-PEG-PLA-NPs or Ab-Mb-VS-PVA-PACE-NPs). Ab-Mb-NPs were flash frozen in liquid nitrogen and stored at −80C until use.

The density of Mb and Ab on the surface of the NPs was indirectly quantified by the concentration in the supernatant from the centrifugation of each conjugation step. The quantification was made by SDS-PAGE. Successful conjugation of the antibody was verified using a cell-based binding assay.

## Mb and Ab conjugation efficiency measured by SDS-PAGE

Each sample was denatured by incubation at 90 °C for 5 min in the presence of 25% (v/v) of LDS sample buffer with DTT at 1:10 (v:v) followed by a centrifugation at $4000 \times g$ for 2 min. The samples were loaded in a 4–12% Bis-tris gel (NuPAGE™ 4–12%, Bis-tris, 1.0 mm, 15 well, ThermoFisher Scientific). The first well was loaded with a standard (SeeBlue™ Plus 2 Pre-stained Protein standard, ThermoFisher Scientific). Four wells were loaded with samples of known concentration to plot a standard curve. The remaining wells were loaded with supernatant from the conjugation steps of unknown concentration. Following the electrophoresis, the gel was stained with Coomassie blue (0.1% Coomassie, 10% acetic acid, 50% methanol, 40% MilliQ water) for 1 h at room temperature under mixing. Subsequently, the gel was destained overnight under mixing. The destained solution (7.5% acetic acid, 10% ethanol, 82.5% MilliQ water) was changed every 30 min for the 2 first hours. Images of the destained gel were taken with a fluorescent scanner (ODYSSEY CLx, LI-COR). The bands intensity was analyzed with Image Studio Lite software (Li-COR Biosciences), using the same ROI and subtracting specific background for each band.

## Cryo-TEM of conjugated NPs

NPs in PBS buffer were centrifuged at $21,000 \times g$ for 15 min at 17 °C. The NPs pellet was redispersed in DI water to reach a final concentration of 5 mg/mL. Quantifoil holey carbon copper grids (Electron Microscopy Sciences) were rendered hydrophilic using a PELCO easiGlow immediately before sample preparation. NPs were vitrified using a FEI Vitrobot cryo plunger (ThermoFisher Scientific) set to 100% humidity by applying 3.5 μL of NPs suspension to the grid followed by automated removal of excess liquid with a filter paper and plunging into liquid ethane. Sample grids were kept in liquid nitrogen at all times and loaded into the ThermoFisher Glacios cryo-TEM (operating at 200 kV) equipped with a Gatan K2 Summit direct detector via the integrated autoloader. Cryo-TEM images were collected under low dose conditions at −4 μm defocus.

## In vitro static cell culture NP binding assay

Human Umbilical Vein Endothelial Cells (HUVECs) were cultured in gelatin- or fibronectin-coated tissue culture plates using M199 medium

supplemented with 20% (v/v) fetal bovine serum, 2% (v/v) L-glutamine, 1% (v/v) penicillin and streptomycin, and 1% (v/v) endothelial cell growth supplement. Porcine Aortic Endothelial Cells (PAECs) were cultured in gelatin- or fibronectin-coated tissue culture plate using Porcine EC Growth Medium supplemented with Porcine EC Growth Supplement. All cells were passaged a maximum of four times. Confluent cells were incubated with NPs at 50 μg/mL in the appropriate media for 2 h, then washed extensively to remove unbound NPs. For imaging purposes, cells were stained with fluorescently labeled anti-CD105 to outline the cells and/or DAPI to mark the nuclei and imaged with a fluorescent microscope (EVOS FL Auto 2, Thermo Scientific Invitrogen) or a confocal microscope (LSM 710 Duo NLO, Zeiss). For quantitative analysis, the cells were lifted with TryPLE, filtered and the intensity of fluorescence of the NPs bound to the cells was measured by flow cytometry (S1000EON, Stratodigm or Cytoflex LX, Beckman Coulter) using fluorescently labeled anti-CD31 (at 0.001 mg/mL) to draw size based gates for endothelial cells with FlowJo v10 software. The auto-fluorescence of HUVECs was subtracted from all values. The gating strategy for HUVECs in flow cytometry analysis is displayed in Supplementary Fig. 16.

## In vitro cell culture under flow

An apparatus of cell culture under flow (Bioflux 200, Fluxion) was used to culture HUVECs in microfluidic channels (24-well plate two inlets one outlet, 0–20 dyne/cm²). First, the channels were coated by flowing fibronectin at 25 μg/mL in PBS for 5 min at 2 dyne/cm² from outlet to inlets followed by 40 min at room temperature without flow. The fibronectin was washed by flowing media from outlet to inlet for 10 min at 1 dyne/cm² and the excess of liquid was removed from all wells. Then, the cells were seeded in the channels by flowing HUVECs at $20 \times 10^6$ cells/mL for 30 s at 1 dyne/cm² from inlet to outlet. The excess of cells in the wells was removed and replace by media in both inlets (250 μL each) and outlet (500 μL) in order to have no flow. After 12 h of incubation (37 °C, 5% CO₂) the cells were confluent. Media was flown for 5 min at 1 dyne/cm² to remove unbound cells and excess of liquid was removed from each well. Finally, NPs at 50 μg/mL in media were flown in the channel from inlet to outlet for 1 h at 0.5 dyne/cm². Pictures were taken during this time using a fluorescent microscope (Olympus Ix71 and Olympus Ix2-UCB) equipped with a 10x objective (Olympus U Plan Fi10x-030PH1 ∞/−) and an exposure time of 0.5 s. After the hour the channel was washed from unbound NPs by flowing phenol free media containing Hoechst 33258 (1 μg/mL) for 5 min at 1 dyne/cm². The channel was imaged under a continuous flow of 0.5 dyne/cm² using the same fluorescent microscope equipped with a 20× objective (Japan U Plan S Apo 20×/0.75 ∞/0.17/FN26.5) and an exposure time of 0.5 s. The fluorescent intensity of the NPs bound to the cells was then quantified from the 20× images (at least 5 images/channel) using custom MATLAB code.

## Ex vivo isolated vessel perfusion

An ex vivo human vessel perfusion system, as described by Lysyy et al., was used to mimic the physiological environment of blood vessels[32]. Briefly, de-identified human umbilical cords were obtained fresh from C-section from the Yale New Haven Hospital. The umbilical artery was carefully dissected from surrounding connective tissue and flushed with cold (4 °C) LR solution. The artery was then cut into ≈8 cm segments in length and mounted in single isolated perfusion chambers by cannulation with flushing needles. They were subsequently perfused in individual closed loops with warm complete M199 media (37 °C) at 2.5 mL/min. A bolus of a 100 μL of NPs at 5 mg/mL was injected in each loop and the perfusion was kept going for 1 h at 37 °C before being stopped. The umbilical artery segments were then recovered and rinsed with clean media. For each, whole-mount en face immunofluorescence confocal microscopy was performed with a 2 mm section

and endothelial cell isolation and flow analysis were performed on the remaining section.

## Endothelial cell isolation and flow cytometric analysis

For gentle endothelial cell isolation, vascular grafts were first washed with warmed PBS. The vessel was then filled with a solution of collagenase II (0.1% in PBS) and incubated for 10 min at 37 °C, and then flushed with ~1 mL of PBS + 1% BSA. The flow through containing endothelial cells was collected in a microcentrifuge tube. Cells were collected from this suspension by centrifugation for 5 min at 3500 × g, and then stained using fluorescently labeled (Alexa Fluor 488, BUV395) anti-human-endoglin (0.005 mg/mL) or an isotype control in PBS + 1% BSA for 50 min. Cells were then washed with PBS + 1% BSA and filtered. The intensity of fluorescence of the NPs bound to the cells was measured by flow cytometry (Cytoflex LX, Beckman Coulter or LSRII, BD Biosciences) and analyzed using FlowJo v10. Cytometric light filters are selected to capture the endothelial cell stain (Endoglin) as well as fluorescence (DiI) from bound NPs. The gating strategy for endothelial cells isolated from human arteries in flow cytometry analysis is displayed in Supplementary Fig. 16.

## Whole-mount en face immunofluorescence confocal microscopy

Whole-mount en face immunostaining and confocal microscopy were used for visualization and evaluation of vascular endothelial layer and NP accumulation and retention. The 2 mm segment of the intact vessel was stained with ULEX-FITC (1 µg/mL) for 60 min at 4 °C. Vessel segments were then washed twice with cold PBS + 1% BSA for 10 min prior staining nuclei with Hoescht 33258 (1:10,000 dilution in PBS + 1% BSA) for 10 min. The samples were placed endothelium-side up on microscope slides, coated with a drop of pure glycerin (glycerol), and then cover slipped. To create as flat a surface as possible, silicone chemical resistant lubricant (Dow Corning, Midland, MI, USA) was applied in a perimeter around the edge of the vessel segment to hold the cover slip in place. Fluorescent images were captured with an LSM 410 spinning-disc confocal microscope and processed using Zen software (Zeiss).

## Ex vivo kidney perfusion

Use of all human kidneys in this study have been approved through the research ethics of New England Donor Services US and Health Research Authority in the UK. Consent for the use of the organs for research was obtained from the donor family by local organ procurement organization representatives before organ retrieval. After in situ flushing of the abdominal organs with cold preservation solution in the donor, kidneys were retrieved and then preserved via static cold storage or hypothermic machine perfusion as described in Supplementary Table 7. Six human kidneys that were declined for transplantation were recruited into the study.

## Inclusion/exclusion criteria

In order to avoid a high level of red blood cell aggregates as previously observed by Tietjen et al. and DiRito et al., we restricted the acceptable time of cold storage of the transplant-declined human kidney enrolled in the study based on the age of the donor[17,41]. This criterion was determined thanks to our previous experiences. The cold storage had to be <10 h for a donor >70 years old, <20 h for a donor between 50 and 70 years old and <30 h for a donor <50 years old. In addition, the organs enrolled had to be suitable for perfusion (e.g. long enough vessel to be plugged onto the machine). For these criterion, we relied upon the opinion of the transplant surgeons and the perfusion team involved in the study. Last, we observed the perfusion parameters and the macroscopic aspect of the organ during the first hour of ex vivo normothermic machine perfusion (EVNMP). Organ were only enrolled for nanoparticles injection if the kidney was perfusing well which was characterized by a homogeneous pink-red color, no spike in resistance, a urine output >100 mL/h/100 g of kidney and a stable blood flow ~50 mL/min/100 g of kidney. With these inclusion/exclusion criteria only one kidney was excluded due to bad perfusion (K1). The other 6 consecutive organs offered fitted all the criteria and were injected with nanoparticles.

## Normothermic machine perfusion

Upon arrival to the laboratory, kidneys were weighed and prepared for perfusion through careful isolation of the renal artery, vein and ureter. The renal artery was cannulated with a 14fr catheter. The ureter was cannulated with a 10fr catheter. Kidneys were flushed with 500 mL of cold Ringer's solution (Baxter Healthcare) prior to perfusion. As previously described, the kidney perfusion circuit utilized pediatric cardiopulmonary bypass technology (Medtronic) and consisted of a centrifugal blood pump (Bio-Pump 560), a heat exchanger (Chalice Medical), a venous reservoir (Medtronic), 1/4-inch polyvinyl chloride tubing, and a Pixie membrane oxygenator (Medtronic)[41]. The hardware included a speed controller and a TX50P flow transducer. The circuit was primed with Ringer's solution and one unit of AB packed red cells (plasma-free) from the New York Blood Center, or blood compatible units of red cells for the kidneys perfused in the UK. 25 mL of 10% mannitol, 8 mg of dexamethasone, 3 mL of heparin (1000 IU/ml), 25 mL of 8.4% sodium bicarbonate, and 10 mL of 5% glucose were added to the perfusate. Ringer's solution was used to replace urine output milliliter-for-milliliter. The perfusate was circulated from the venous reservoir through the centrifugal pump at 1450 rpm into the membrane oxygenator, where it was oxygenated and also warmed to 38 °C. It then flowed through the arterial limb of the circuit to the renal artery. Venous return from the renal vein was fed back into the reservoir. Kidneys were placed on the circuit and perfused for an initial assessment period of 60 min. If the perfusionist deemed that the kidney was stable and functioning well at the end of that 60-min period, the experiment would proceed with the NPs injection. A bolus of 10 mL of NPs suspension containing non targeted NPs (DiO loaded) and Ab-Mb-NPs (anti-CD31 or Isotype, DiI loaded) both at 5 mg/mL would be delivered into the arterial port of the perfusion circuit reaching a final circulating concentration of 50 µg/mL in the perfusate. The perfusionist was blinded to what NPs they were delivering in each kidney for the four last kidneys (two pairs) enrolled. The perfusion was continued for an additional 4 h after NPs injection.

## Sample collection and processing

Renal blood flow (RBF) and total urine output was continuously monitored and registered throughout the NMP procedure. Perfusate and urine samples collected throughout the perfusion were snap frozen and stored at −80 °C. Wedge biopsies were collected prior to the start of perfusion and before NPs injection at 1 h of NMP. At the end of the 5-h procedure kidneys were flushed with LR, bisected and three biopsies spanning the cortex and medulla were collected (one in the upper part, one in the middle and one in the lower part). Biopsies were snap-frozen in liquid nitrogen upon collection and stored at −80 °C until further analysis. From each post perfusion biopsy and from the biopsy before NPs, three tissue sections 100 µm apart in depth were stained with *Ulex europaeus* Agglutinin I (ULEX, a lectin which can be used to identify endothelial cells), and whole sections of tissue (Fig. 5a, b) were imaged at 20x resolution and tiled (200 to 350 images per section) using a fluorescent microscope EVOS FL Auto 2 (Thermo Scientific Invitrogen). Filter cubes with the following parameters were used for each channel: blue channel with 357/44 nm Excitation; 447/60 nm Emission, green channel with 482/25 nm Excitation; 524/24 nm Emission, red channel with 531/40 nm Excitation; 593/40 nm Emission, and far red with 628/40 nm Excitation; 692/40 nm Emission. A consistent laser power (50% laser power) and exposure were used for all image collection (0.2 s exposure for green and 0.15 s exposure for red, 0.05 s exposure for far red).

## Image analysis

Images were analyzed using custom MATLAB code which is available upon request and is described in Supplementary Fig. 8. Briefly, the background signal in both the green channel and the red channel was measured from biopsy taken before NPs were introduced for each kidney. Using an average of the values obtained from ~900 images, a value was calculated that would make 98% of the image black if subtracted from each pixel. This value was around 400 for green images and 150 for red images. Next, a vascular binary mask was built based on positive ULEX staining in the far red channel. Briefly, each image in the far red channel was analyzed using an adjusted Otsu's method to determine a threshold that when applied, resulted in an accurate area representation of the vasculature in each image. This value varied between 600 and 1200 based on the particular kidney and on the region of tissue (cortex vs medulla). The determined threshold value was applied to the far red image, resulting in a binary mask (value of 1 where vasculature is present, value of 0 where it is not). This mask was applied by multiplication to each of the green and red channel images, to remove signal from tissue regions not contained inside the vasculature. Next, an inverse binary mask was built from positive green (nonspecific) NP signal using a threshold of 900, which was predetermined to accurately represent green NP accumulation area. This mask (value of 1 where no NPs were found, value of 0 where green NPs were found) was applied by multiplication to the red channel image, to remove areas of green and red NP colocalization. Finally, the corrected red channel image was binarized (using a threshold of 400, and the sum of nonzero elements per image was calculated. The cumulative sum of all nonzero elements in the red channel per slice (~300 images) was then divided by the cumulative sum of nonzero elements in the vascular channel minus the cumulative sum of nonzero elements in the green channel (vascular mask applied) per slice to obtain a percent of available vascular area covered by NPs.

## Statistical analyses

The results of the experiments are expressed as the means ± the standard deviation. Statistical analyses were noted were performed by ANOVA and a post hoc Tukey's Test where appropriate for comparison between groups using Prism 8.0 (GraphPad Software, Inc.). In other cases, where noted, statistical analysis was performed with multiple $T$-tests with a Bonferoni correction using Prism 8.0 (GraphPad Software, Inc.). A value of $p < 0.05$ was considered statistically significant. Exact $p$ values can be found in the source data file.

## Reporting summary

Further information on research design is available in the Nature Research Reporting Summary linked to this article.

## Data availability

The data generated in this study are provided in the Supplementary Information and Source Data files. Raw images are available upon request. Source data is available for Figs. 2–5 and Supplementary Figs. 2, 13, 14 in the associated source data file. Source data are provided with this paper.

## Code availability

All custom MATLAB code is also available upon request.

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

## Acknowledgements

L.G.B. was supported by NIH NRSA training grant (T32DK007276) and a K99/R00 Pathway to Independence award (K99 HL157552). This work was supported by grants from National Institutes of Health (U01 AI132895 to W.M.S. and J.S.P.; U01 AI145965 to W.M.S., R01 CA194864 to S.K.), the American Association for the Study of Liver Disease (AASLD; G.T.T.) and the National Institute for Health and Care Research (NIHR) Blood and Transplant Research Unit in Organ Donation and Transplantation (NIHR203332, S.A.H), a partnership between NHS Blood and Transplant, University of Cambridge and Newcastle University. The views expressed are those of the author(s) and not necessarily those of the NIHR, NHS Blood and Transplant or the Department of Health and Social Care. Cryo-EM data were collected at the Yale CryoEM Resource that is funded in part through the NIH grant S10OD023603. The authors would like to sincerely thank all the donors and their families. Without their generous gifts, this study could not have been conducted. The authors also thank New England Donor Services for their assistance in obtaining organs for research.

## Author contributions

C.A. and L.G.B. contributed equally to the targeted NPs design, experimental design and execution, data analysis, interpretation, and presentation, and manuscript composition. The design, production and validation of the Mb was conducted by A.K., E.D., G.K., T.H., and S.K. Organ and vessel perfusion experiments were also conducted by J.D., T.L., C.E., O.R., J.L., D.H., S.A.H., and M.L.N. Organ biopsies were pathologically evaluated by S.P. N.P. description experiments were also conducted by J.G. and K.Z. Data presentation was contributed to by L.H. and A.F. L.H. also contributed to data analysis and manuscript composition. J.S.P., W.M.S., S.K., and G.T. contributed to the targeted N.P. design, experimental design, data interpretation, and manuscript composition.

## Competing interests

A.K. and S.K. are listed as inventors on issued and pending patents on the monobody technology filed by the University of Chicago and Novartis AG (US Patent 9512199 B2 and related pending applications). G.T., S.K., W.M.S, C.A., J.S.P., A.K., and L.G.B. are listed as inventors on pending patent using the Mb adapter to build targeted NPs filed by Yale University and New York University (US Patent 20210206879 A1). G.T.T., S.K., and J.D. are co-founders and equity stake holders in Revalia Bio Inc. S.K. was a SAB member and held equity in and received consulting fees from Black Diamond Therapeutics, and has received research support from Argenx BVBA, Black Diamond Therapeutics and Puretech Health. The remaining authors declare no competing interests.
