## [Peer Review File · Nature Communications]

Monobody adapter for functional antibody display on nanoparticles for adaptable targeted delivery applicationsREVIEWER COMMENTS

Reviewer #1 (Remarks to the Author):

Tietjen et al present a novel approaches to functionalize the surface of nanoparticles with antibody-based targeting ligands. They developed fibronectin-based monobody adapters, which via c-terminal cysteine moieties are grafted onto the surface of NP, and which via binding to the Fc portion allow for properly oriented antibody modification of the NP surface. The work is interesting, the manuscript well-written and the figures scholarly presented. The key question is whether the monobody adapter technology represents a sufficiently large advance in fundamental understanding and/or technological capability to justify publication in Nat Commun. There are also multiple aspects that can be addressed to further improve the quality of the manuscript.

Most important issues:

- The key advance reported is the monobody adapter technology, not EC targeting in kidney transplants ex vivo (that was covered in Ref 9). It would be good if the authors could better explain how the monobody adapter approach extends / improves current NP surface modification methodology. How does it compare to "standard" biotechnology-based methods for antibody premodification? Or to sortase-based surface modification? It would be nice to experimentally demonstrate added value over such alternative approaches.
- I can follow the general direction of targeted EC in transplant settings, but would like to learn from the authors in which concrete situation they see added value for this approach. With which drug? For which indication (preventing transplant rejection? How can that be achieved? Are then rejected kidneys are good example for performing such studies?). For the broader context of the work, it would be nice to add some sentences on this in the introduction and/or discussion.
- Figure 1: Panels C and D are mixed up in the legend. Also, panels D,E,F show a relatively large discrepancy in size between DLS and TEM (of 50-60 nm); how come? Furthermore, in panel C, it would be nice to document with the number of antibodies on the surface of NP (rather that the weight in ug Ab / mg NP; can be easily calculated).
- Related to last comment above: given that the amount of Ab on the NP surface is "only" doubled, how to explain the large increases in EC targeting? I can barely imagine that improving orientation helps to achieve 50-500-fold enhancement in binding. Interpreting this better would profit from knowing how many Ab are on the surface of each NP. Furthermore: why is the 1000-fold difference only achieved under flow in vitro (1000-fold), and so much larger than under static in vitro (15-fold) and flow ex vivo in umbilical arteries (10-fold)?
- Fig 5: The authors claim that as compared to their previous study (Ref 9), use of the monobody adapter improves area coverage in the glomeruli from 5% (Fig. S8) for standard CD31-NP to 72% for CD31-Mb-NP (Fig 5H). However, looking at Fig. S8, this claim is not correct. Fig. S8E shows area coverage of 20%. This does confirm the added value of the monobody adapters, but also shows that the advance over standard NP surface modification is not that large (only a factor 3.5). As indicated above, a direct and quantitative head-to-head comparison would be preferred to resolve this issue.

Additional issues:

- Figure 2: Header "cell culture under flow" above panels A-D is incorrect, should be "under static conditions"
- Figure 3E and Figure 2K show exactly the same data for CD31-Mb-NPs.
- Figure 5: Isn't it possible to test isotype NP first and then specific NP in the same kidney? Or both at the same time with different colors? The same would hold true for head-to-head comparing standard Ab-NP versus Ab-Mb-NP. This would enable more direct comparison.

- Figure 5: The overlap between vascular cells in cyan and NP is not obvious. How come? Can this be quantified via microscopy and/or confirmed with FACS?

- The authors on multiple occasions wrongly assume that active ligand targeting reduced off-target localization and toxicity. They state that EC-specific therapy can help to avoid off-target accumulation in liver and spleen. I don't think there is any evidence for this, since adding a specific ligand will not reduce unspecific phagocyte uptake. Moreover, even if the target site dose upon ligand targeting goes from 1 to 10 %ID, the vast majority will still end up in phagocytes in liver and spleen. Unless, of course, the particles bind to vascular endothelium everywhere. Which I cannot imagine is what the authors are referring / aiming to. They should therefore remove the claims on ligand-mediated avoidance of off-targeting (or provide much more (con)text explaining the rationale behind this).

- Last sentence of both abstract and manuscript: Why would this technology "accelerate/hasten the preclinical development and clinical translation"? I think that depends on many (other) things, and don't see how a refinement strategy helps to accelerate the process...

- What is known about the immunogenicity of the monobody adapters?

- Does the technology enable (more controlled and/or more facile) dual or triple targeting? Does that make it potentially stand out among current NP surface modification technologies?

Reviewer #2 (Remarks to the Author):

The authors present the use of a monobody as a chemical linker to immobilize antibody on the surface of nanoparticles. The Mb facilitates covalent attachment, proper orientation, and high loading density to the NP. The authors thoroughly characterize the conjugation and demonstrate the specificity of the Mb for mIgG-Fc binding. In addition, the authors compare this novel conjugation strategy to conventional EDC/NHS coupling chemistry. The results are quite impressive and address a critically needed method to improve conjugation of antibodies to NPs for highly functional bioconjugates that is also adaptable to many systems (unlike protein engineering). Most impressively, the authors show substantial improvement in conjugate binding in systems that are more appropriate models of clinical samples compared to the EDC/NHS prepared conjugates. Overall, the work is well-presented, highly significant, and scientifically sound. Several critiques are provided below, and minor revisions are recommended prior to publication.

1. The authors note that EDC/NHS prepared conjugates work in simple model systems (although less effective than Mb-mediated conjugation). However, the Mb conjugates provide significantly greater binding (>1000-fold) compared to EDC/NHC conjugates in more realistic setting. The authors attribute this improvement to better orientation and greater antibody loading when using Mb to prepare conjugates. It seems unlikely that these two factors alone can justify a 1000-fold difference in binding. If conventional conjugation provides 50% of the antibody loading and assume only ~30% are active/oriented, then the conventional conjugates still have 15% of the functional antibodies that the Mb conjugates present. Is it possible that the EDC/NHS linkage was not stable or effective (not covalent but electrostatic binding of antibody)? Alternatively, is it possible that multiple antibodies are required to bind the cells to tightly anchor the NP to the cell surface (multi-dentate binding), which would require a higher density ordered array of Ab on the NP surface? The data are compelling and sound, but it would be great to better explain the substantial improvement in NP binding for realistic samples.

2. If the monobody is designed to universally bind mIgG1, will other mIgG1 molecules in a biological sample compete/displace the pre-adsorbed mIgG1 on the conjugate? This could be an issue when the Mb is redesigned to bind humanized antibodies and the conjugates are introduced

into human samples which may contain many other hIgG proteins that could potentially result in antibody exchange and loss of function.

3. The Ab and Mb loading on the NPs was measured using SDS-PAGE. More conventionally, total protein assays like BCA are used to quantify excess antibody in the supernatant. Can the authors comment/justify the choice of SDS-PAGE for quantitation and/or show calibration in supporting information? The precision and analytical sensitivity of this method would be valuable to show to the readers.

4. Can the authors comment on how the Mb compares to protein A/G-mediated conjugation that has previously been used to bind IgG-Fc and facilitate controlled orientation of antibodies bound to NPs?

5. In Figure S1, mIgG1-Fc bound better than whole mIgG1. However, what if two full length mIgG1 molecules completed for FCM101 binding? (similar to comment 2)

6. Figure S1 caption (b) "Yeast cells displaying FCM101 were first mixed with FCM101 (50 nM)," Should this be mixed with mIgG1-Fc (50 nm)?"

7. There are 9 Mb for every 1 Ab on the NP surface. Why not 1:1? Is this a size argument (antibodies are bigger and take up more surface area) or are some Mbs inactive, improperly oriented, etc.? Based on the surface area of NP and size of an Ab and Mb, what is theoretical Ab and Mb loading?

8. Figure 2 heading for A-D "Cell culture under flow" should be "Static Cell Culture".

9. Caption for Figure 3 (F-I) reads as if all samples are PAECs that are incubated with different labels; however, Figure and text suggests samples differ (PAECs and HUVECs) while labels are the same.

Reviewer #3 (Remarks to the Author):

Overall the paper is concerned with optimising a previous study where mAbs were conjugated to nanoparticles. Monobodies are introduced without the context of previous work such notably the adnectin-anti-VEGFR2 paper (first use of introducing a cysteine for PEG conjugation), or the centyrin work (inserted a cysteine at every position to find the best site for conjugation).

The authors have created a nanoparticle approach that ensures conjugated antibodies are outward-facing. They demonstrate:

- Using monobodies as an adaptor to connect anti-ICAM2 mAbs with PLA-PEG nanoparticles, which increases nanoparticle binding to target epithelial cells over NPs with the mAb directly conjugated
- Then swapping the anti-ICAM2 mAb for an anti-CD31 mAb and demonstrating specific binding to the new target
- Then swapping the PLA-PEG to a PACE-PVA nanoparticle
- Then flushing a human kidney (rejected for transplant) with anti-CD31-Mb-NPs and demonstrating specific binding

Generally, I find this technology to be overly complicated:

1. I don't understand the significance of monobodies in this technology:

- The full antibodies could be conjugated via cysteine chemistry (e.g. <https://www.sciencedirect.com/science/article/abs/pii/S0168365920300547> from 2020),
- or cut down to the ScFvs and conjugated via cysteines (e.g. <https://pubs.acs.org/doi/abs/10.1021/ja0555668> from 2006)
- The monobody could be alternatively replaced with a protein G, FcRn or other Fc-carrying protein

2. In the case that the monobody is somehow essential, why attach an antibody?

- The antibody format used is from a mouse, which will cause further immunogenicity in a transplanted organ
- The authors already argue that generating a new monobody with binding to different human IgG types is not a challenge, so why not just raise a monobody directly against your target of interest? o i.e. their words "Just as human Abs can be developed using many modern technologies, Mbs to human Fc can be readily developed following the well-established pipeline^{12, 21}."
- Of note, inserting a cysteine into monobodies for conjugation is also not a novel approach (e.g. <https://pubmed.ncbi.nlm.nih.gov/27737926/> or <https://www.ncbi.nlm.nih.gov/pmc/articles/PMC2840239/>) which have not been mentioned despite being core works in the field.

Overall, interesting study but is effectively just an optimisation of their previous work (<https://stm.sciencemag.org/content/9/418/eaam6764.short>), and uses a technology that adds the complexity of an adaptor, which could be avoided by instead optimising the initial IgG conjugation chemistry. Overall, I think it's an interesting, detailed and carefully performed study but maybe not groundbreaking.

Reviewer #1 (Remarks to the Author):

Tietjen et al present a novel approaches to functionalize the surface of nanoparticles with antibody-based targeting ligands. They developed fibronectin-based monobody adapters, which via c-terminal cysteine moieties are grafted onto the surface of NP, and which via binding to the Fc portion allow for properly oriented antibody modification of the NP surface. The work is interesting, the manuscript well-written and the figures scholarly presented. The key question is whether the monobody adapter technology represents a sufficiently large advance in fundamental understanding and/or technological capability to justify publication in Nat Commun. There are also multiple aspects that can be addressed to further improve the quality of the manuscript.

We thank the reviewer for their complementary remarks. We'd like to highlight that the key advance here is the use of an adapter technology that provides two important functions: versatility in coupling to antibodies (Abs) with different binding specificities and substantially improving the efficiency of NP binding through the noncovalent and oriented coupling of Abs to the NP surface. While it is possible that other adapter systems could be engineered to accomplish these same outcomes, to the best of our knowledge, none have yet been presented that achieve both functions without additional steps.

Most important issues:

1. – The key advance reported is the monobody adapter technology, not EC targeting in kidney transplants ex vivo (that was covered in Ref 9). It would be good if the authors could better explain how the monobody adapter approach extends / improves current NP surface modification methodology. How does it compare to “standard” biotechnology-based methods for antibody premodification? Or to sortase-based surface modification? It would be nice to experimentally demonstrate added value over such alternative approaches.

The principal improvement over methods that require premodification to improve orientation (such as sortase- or biotinylation-based conjugation) is that no genetic or chemical modification of the Ab is needed with the monobody approach. This modularity greatly simplifies the early development phase of studies (where an optimal target is not yet identified) by allowing the testing of many different unmodified Abs with the intended NP. We suggest that the experiments shown in Figure 3, where the Ab is easily switched between 3 off-the-shelf Abs without any optimization or modification steps, to either the Ab or NP, demonstrate this significant value and point to many potential impacts. We have added text to the Introduction and Discussion to better clarify this point.

Introduction (**BOLD** indicates changes, in this and other responses):

“We posited that the lack of efficiency of our prior targeted NPs in human organ delivery was a result of the method by which the Abs were associated to the NP surface. Abs were conjugated to the surface of NPs through covalent coupling of the Ab free amines via the standard EDC-NHS chemistry¹⁷. This approach is widely used because of its simplicity and ease of use¹⁸. However, well described limitations of this method include random orientation of the coupled Abs and potential modification of the **lysine residues crucial for antigen binding**. Both of these limitations lead to diminished binding potency due to a fraction of the Ab binding sites being unavailable¹⁸⁻²¹. **To partially mitigate this problem some have used Fc-binding proteins – such as protein A/G or FcRn to link the Ab and the NPs. These approaches only allow a partial control of the orientation as these adapter proteins are themselves randomly oriented at the surface of the NPs via EDC-NHS chemistry²²⁻²⁴. This might negatively impact the surface density of the targeting Ab if not all adapters are available for binding.**”

As an alternative, targeting molecules can be engineered to support site-specific conjugation in a manner that avoids obscuring the sites of antigen binding (**click chemistry, biotin-binding proteins, enzyme-based bioconjugation, and others**). **These alternative strategies are comprehensively reviewed by Sivaram et al¹⁸. While effective, these require modification of each Ab, which is labor intensive and costly, diminishing the ability to rapidly screen antibodies targeting different antigens. During the early development phase of a targeted NP it is essential to identify the optimal antigen and targeting reagent for the given species, disease, organ, and cell. A method to control Ab orientation without having to re-engineer each individual targeting molecule under consideration would be impactful. Here we describe a new approach that combines simplicity, adaptability and efficiency by using unmodified antibodies as the targeting molecule.**"

"We have also demonstrated the versatility of this approach by showing how the Mb adapter enables easy exchange of both the NP and targeting Ab without sacrificing binding efficiency or adding optimization steps. In other words, **Ab-Mb-NPs are composed of three distinct blocks: 1) the drug-encapsulating nanoparticle, 2) the Mb adaptor and 3) the targeting Ab. The Mb block serves as a universal adaptor that links diverse combinations of NP and targeting Ab blocks that can be chosen to fit both the therapeutic need and corresponding target.**

Discussion:

"Ab-Mb-NPs can also target different molecules or species by simply swapping any available Ab of the same isotype or combining multiple of these Abs to hit different targets at once. **This provides an improvement over the strategies that require Ab modification.** The exchange of the targeting Ab could be used to facilitate the screening and discovery of the optimal antigen to target for a particular tissue, or could be motivated by a transition between pre-clinical models (e.g. porcine to human organs). In addition to the versatility of the targeting molecule, our design of the FCM101 construct with a single Cys residue enables its conjugation to a variety of NPs presenting a thiol reactive group on their surface. This approach grants the ability to rapidly screen for the best vehicle for a particular therapeutic without further modifications to the targeting strategy and expands the possible applications. **In the end, this approach can be thought of as a building-block design where each critical component – therapeutic agent, NP, and antibody - can be switched without modifying the other blocks. Thus, we envision that our approach will facilitate more comprehensive screening for the optimal combination of these three components.**"

2. - I can follow the general direction of targeted EC in transplant settings, but would like to learn from the authors in which concrete situation they see added value for this approach. With which drug? For which indication (preventing transplant rejection? How can that be achieved? Are then rejected kidneys are good example for performing such studies?). For the broader context of the work, it would be nice to add some sentences on this in the introduction and/or discussion.

We thank the reviewer for the suggestion to discuss potential clinical impact of the work. As a clarification, we experiment upon "declined" kidneys (i.e. not used for transplantation) not "rejected" kidneys following transplantation. Major reasons such organs are declined include extended ischemic time or donor health, conditions that predispose the recipient to complications like graft rejection and/or organ loss.

Targeting therapeutics to graft ECs during ex vivo NMP can reduce the host immune system's rejection of these organs. For example, we can reduce class II MHC molecule expression with siRNA to CIITA⁸ or NLRP3 inflammasome assembly with MCC950 in graft ECs¹⁵, both treatments that reduce rejection in transplant

models. As such, these organs are an ideal translational model of those settings where our therapies would be most beneficial. We have added text to the introduction and the discussion describing our envisioned applications for targeted NPs in EVNMP.

Introduction:

“The risk of dysfunctional inflammation is greater in marginal organs – organs from older and less healthy donors - and contributes to these organs being declined for transplant more frequently due to higher risk of post-transplant complications. Following organ recovery and preservation, ECs play a critical role in post-transplant pathologies associated with dysfunctional inflammation. A single dose of therapeutics delivered in the form of vascular-targeted NPs during EVNMP has the potential **to reduce the immunogenicity of the graft at the EC level and** to provide several weeks of protection against dysfunctional inflammation during the post-transplant period when the organ is in its most vulnerable state⁸. **Reduced endothelial activation has been demonstrated by treatment with anti-inflammatory molecules that inhibit NFκB⁹⁻¹¹, mTOR¹², and complement¹³⁻¹⁵ pathways as well as by siRNA-specific knockdown of inflammatory proteins (e.g., MHC⁸ and IL-15¹⁶).** Thus, effective delivery of therapeutics via vascular targeted NPs during EVNMP can both circumvent the challenge of organ-specific targeting and administer impactful therapies, **which could expand the pool of transplantable organs.”**

Discussion:

“Ab-Mb-NPs also resulted in excellent vascular area coverage when perfused in human kidneys during EVNMP. This result is encouraging for future applications of therapeutic delivery **to marginal organs, such as the kidneys enrolled in this study.** Targeted NPs to treat marginal organs prior to transplant could reduce the risk of **dysfunctional endothelial cell inflammation and associated recruitment of immune cells**, rendering those organs safer to transplant and thereby increasing the number of transplantable organs.”

- Figure 1: Panels C and D are mixed up in the legend. Also, panels D,E,F show a relatively large discrepancy in size between DLS and TEM (of 50-60 nm); how come? Furthermore, in panel C, it would be nice to document with the number of antibodies on the surface of NP (rather than the weight in ug Ab / mg NP; can be easily calculated).

We apologize for the mislabeling and have corrected the legend. We also added a second axis to Fig 1C to represent the number of Ab per NP calculated by assuming a NP of 100 nm diameter and 1.25 g/cm³ density.

Regarding the size discrepancy, the size measured by DLS corresponds to an average hydrodynamic diameter which includes the NPs hydration layer – a factor that is usually enhanced in the presence of PEG on the surface. The reported value is also intensity-weighted which overemphasizes larger particles in the calculation. In contrast, the TEM shows the electron dense core of the NPs without the hydration layer. These differences commonly result in smaller size measurements by TEM than DLS. Other reports in the literature compare these two techniques; the ~50-60 nm difference in size measured here is consistent with the differences observed by others [1]. In addition, the images of cryo-TEM were chosen to best display the organization of the Ab on the surface of the NPs and not necessarily to be representative of the average size of the sample. In fact, the Pdl of 0.15 obtained by DLS of these samples corresponds to a standard deviation of roughly 60 nm which would include the size of the NPs shown in the cryo-TEM image. The caption was modified to clarify this point.

1. Maguire, C.M., Rosslein, M., Wick, P. & Prina-Mello, A. Characterisation of particles in solution - a perspective on light scattering and comparative technologies. *Sci Technol Adv Mater* 19, 732-745 (2018).

- Related to last comment above: given that the amount of Ab on the NP surface is “only” doubled, how to explain the large increases in EC targeting? I can barely imagine that improving orientation helps to achieve 50-500-fold enhancement in binding. Interpreting this better would profit from knowing how many Ab are on the surface of each NP. Furthermore: why is the 1000-fold difference only achieved under flow in vitro (1000-fold), and so much larger than under static in vitro (15-fold) and flow ex vivo in umbilical arteries (10-fold)?

This result was surprising to us as well and validates the effectiveness of our new approach. Reports in the literature have shown a nonlinear (exponential) increase in NP binding as the density of a targeting Ab increases above a certain threshold. This is likely due to increases in avidity due to multivalent binding, since small changes in multivalency can produce orders of magnitude increases in binding kinetic constants (modeled³⁴, and reviewed³⁵). Our data suggest that it is the combination of proper antibody orientation and greater total Abs per particle (doubled) that results in many more active Abs on the NP surface, and this in turn drives the multiple orders of magnitude improvement over the NPs prepared using conventional chemical coupling methods.

In addition, the difference in fold-change observed in the different experiments under flow and in static in vitro are affected by the different experimental conditions. Of most influence, the experiment conducted on cells cultured under flow (Fig 2E-G) involves a constant replenishment of NPs, maintaining a concentration of 50ug/mL for the full 1 hr of the experiment. In contrast, in static cell culture there is a single administration of 50ug/mL which likely results in a diffusion-limited binding as the NPs are depleted in the unstirred layer at the cellular interface. Finally, the ex vivo vessel perfusion also uses a single administration of a concentrated bolus of NPs (resulting in 50ug/mL in the perfusate), and is conducted using an arterial segment. Prior studies have shown differences in the ECs that line the umbilical artery in situ from HUVECs in culture that also may play a role in NP binding [1]. We have modified the text in the results section to describe these differences more clearly.

1. Liu, M., Kluger, M.S., D'Alessio, A., Garcia-Cardena, G. & Pober, J.S. Regulation of arterial-venous differences in tumor necrosis factor responsiveness of endothelial cells by anatomic context. *Am J Pathol* 172, 1088-99 (2008).

Results section:

“The difference in fold change between the three experiments presented here can be explained by differences in the experimental setting. For example, in cell culture under flow the NPs are constantly replenished as compared with a single administration in static cell culture where binding may be diffusion limited as the NPs are depleted in the un-stirred layer at the cellular interface. In the ex vivo vessel perfusion, the binding environment is much more complex than ones involving monolayers of cultured cells and introduces new challenges to binding that are not accounted for in cell culture systems. Together, our new Ab immobilization method dramatically increased targeted delivery of NPs compared with the conventional direct coupling in the multiple systems tested. We hypothesize that this improvement is due to substantial increases in the surface density of active Abs on the NPs (increased in number and by a more functional orientation), which has increased the instances of multivalent interactions between cell-surface antigen and NPs. Increased multivalency is expected to increase binding efficiency exponentially^{34,35}.”

- Fig 5: The authors claim that as compared to their previous study (Ref 9), use of the monobody adapter improves area coverage in the glomeruli from 5% (Fig. S8) for standard CD31-NP to 72% for CD31-Mb-NP (Fig 5H). However, looking at Fig. S8, this claim is not correct. Fig. S8E shows area coverage of 20%. This does confirm the added value of the monobody adapters, but also shows that the advance over standard NP surface modification is not that large (only a factor 3.5). As indicated above, a direct and quantitative head-to-head comparison would be preferred to resolve this issue.

We are sorry about this mistake in the text and we have changed it for the appropriate value. The increase in area of coverage from 20% to 72% is considerable since this is a percent of coverage (maximum of 100%) and not a measure of total NP binding as was used in the in vitro assays. Additionally, 72% approaches the maximum we can achieve in reality due to the areas occluded by red blood cell aggregates that were removed to only account for specific binding. In order to present this point more clearly, we have now modified the graphs in figure 5 and in figure S15 to present the area of *available* vasculature covered with red NPs (i.e. the area covered with red NPs normalized to the vasculature area without aggregates). We also added Supplementary Figure 12 that shows the data previously displayed in Figure 5 (total vascular area covered with red NPs) as well as the area covered with aggregates. Together these figures show that Ab-Mb-NPs achieved near 100% coverage if we account for the NPs in aggregates, suggesting that clearing of these aggregates from the organ before the NP injection would result in near 100% coverage of vascular area. This argument is supported by specifically looking at kidney 5, which shows very few aggregates (~6% of vascular area), where NP coverage reached ~90% of vascular area (Fig. S12). Furthermore, we would like to draw attention to the more substantial difference in microvascular coverage between the CD31-NPs and the C31-Mb-NPs, which increased by a factor of almost 40. As opposed to the glomeruli, microvasculature doesn't benefit from the smaller size and tortuosity of the glomerular microvessels, or the local increase in NP concentration, and which could all increase the probability of a binding event, making binding in the microvessels all the more challenging. Taken all together, the new analysis supports substantial increases in coverage achieved with Ab-Mb-NPs.

We agree that a head to head comparison is a better way to present this issue. Therefore, we have added new experiments that compare the binding of CD31-Mb-NPs to CD31-NPs (NPs which have been conjugated to anti-CD31 via EDC-NHS chemistry). These results can now be seen in Supplementary Figure 13 and 14. The vascular pixel area was significantly higher with CD31-Mb-NPs than CD31-NPs in both the microvessels and in the glomeruli. Specifically, microvascular pixel coverage was 17% vs 4% in the pair and 16% vs 4% in the single. Glomerular pixel coverage was 92% vs 19% in the pair, and 56% vs 11% in the single.

The benefit from the Mb adaptor is preserved in these new experiments but is at the lower end of the binding advantage we observed in the initial series of experiments. We believe this is due to the notably worse quality of kidneys MbK11-12 (the new paired experiment), which both prevented optimal perfusion and reduced NPs available for binding, as the dose is partially lost to clogged vessels. We note significant damage in kidney 11 and 12 (pair) (including sclerosed glomeruli, tubular drop out, and fibrosis), and include a panel of H&E-stained images with a pathology score in Supplementary Figure 8 to support this view. Additionally, analysis of the internal control in the pair of kidneys (untargeted NP accumulation) revealed that MbK11 had 29% aggregated area, which is 17% higher than any previous kidney. In the past, we would not have used kidneys in such poor shape in our experiments because such organs would never be considered for use in transplantation. However, the availability of kidneys for research has been significantly impacted as healthcare systems respond to COVID-19. After waiting almost one year to conduct these experiments, we elected to relax the inclusion criteria so that we could respond to your

critique by conducting these new experiments. Finally, the coverage of NPs in MbK13 (single kidney with both NP sets) should be considered independently from other experiments, since we do not know the effect of competition between the two types of NPs. Most importantly, despite the poor quality of these kidneys, there is still significantly improved vascular binding and coverage with CD31-Mb-NPs over CD31-NPs.

Additional issues:

- Figure 2: Header “cell culture under flow” above panels A-D is incorrect, should be “under static conditions”

Thank you for noticing, we have corrected the figure.

- Figure 3E and Figure 2K show exactly the same data for CD31-Mb-NPs.

We agree, and have acknowledged this in the figure captions, but we feel that showing these data twice is important for the comparison to the other groups in both figures.

- Figure 5: Isn't it possible to test isotype NP first and then specific NP in the same kidney? Or both at the same time with different colors? The same would hold true for head-to-head comparing standard Ab-NP versus Ab-Mb-NP. This would enable more direct comparison.

Serial perfusion would complicate the analysis since flow changes over time on the pump and aggregates form during perfusion. As noted above, we have added new experiments that confirm the advantages of Ab-Mb-NPs vs. Ab-NPs.

- Figure 5: The overlap between vascular cells in cyan and NP is not obvious. How come? Can this be quantified via microscopy and/or confirmed with FACS?

Due to the many colors that are superimposed in Figure 5, it is indeed difficult to visualize the overlap. To make this clearer we added Supplementary Figure 10 which shows single color images of the vascular stain and of the NP signal, as well as an overlay of only these two signals. This figure shows that the vast majority of the NPs are co-localized with vascular cells. A quantification of this concurrence (i.e. how many cells have both ulex and NPs) is already included in our analysis in Figure 5, which show NPs signal only where they colocalize by pixel with the vascular cell signal (cyan).

- The authors on multiple occasions wrongly assume that active ligand targeting reduced off-target localization and toxicity. They state that EC-specific therapy can help to avoid off-target accumulation in liver and spleen. I don't think there is any evidence for this, since adding a specific ligand will not reduce unspecific phagocyte uptake. Moreover, even if the target site dose upon ligand targeting goes from 1 to 10 %ID, the vast majority will still end up in phagocytes in liver and spleen. Unless, of course, the particles bind to vascular endothelium everywhere. Which I cannot imagine is what the authors are referring / aiming to. They should therefore remove the claims on ligand-mediated avoidance of off-targeting (or provide much more (con)text explaining the rationale behind this).

We fully agree with this important point, and we are sorry that our language in the manuscript was not clear. We have changed the language in the abstract and the introduction in two ways: 1) to emphasize the fact that we specifically are exploiting ex vivo machine perfusion to avoid off-target accumulation of

the NPs because the organ is isolated from the rest of the body and by consequence has no spleen or liver to take the NPs and 2) to clarify that targeting is intended to enhance retention not to reduce off target binding.

- Last sentence of both abstract and manuscript: Why would this technology “accelerate/hasten the preclinical development and clinical translation”? I think that depends on many (other) things, and don’t see how a refinement strategy helps to accelerate the process...

A strength of this technology is its modularity, which we describe as a building block design in our response to your first comment. The development phase might be simplified and thus accelerated by allowing the use of unmodified Abs. As R&D and preclinical research progress between diseases, and from cells to animals to human, new targets can be easily assessed without investing time and money into special design and production of Ab or derivatives as shown to some extent in Figure 3. In addition, once a targeting strategy is in place, the drug-nanoparticle can be readily swapped for an alternate combination to treat a different pathology in the same cell type. We demonstrate this in Figure 4 where the same targeting strategy (i.e. same Ab and Mb) is successfully applied to PACE polymer which is optimized for nucleic acid delivery. While we agree many factors play a role in the process of developing nanotherapeutics, we still contend that modularity can accelerate the process. We have adjusted the claim to represent our view more accurately, specifically by changing the language to “simplify and possibly accelerate”, in addition to changes described in response to your first comment.

- What is known about the immunogenicity of the monobody adapters?

We have not examined the immunogenicity of this particular monobody, FMC101, in humans or animals. However, a clinical trial of a related molecule (Adnectin) showed that the immunogenicity is generally low and anti-drug antibodies developed only to mutated segments [1], in an analogous manner to anti-idiotypic antibodies developed for fully human therapeutic antibodies. FMC101 has approximately half as many mutated residues as typical human antibodies that have mutated residues in the VL and VH domains. Based on these considerations, we suspect that FCM101 is no more, if not less, immunogenic than typical therapeutic antibodies. Moreover, as our primary proposed applications will be ex vivo, immunogenicity is less likely to be of concern. We added a sentence in the discussion to clarify this point.

1. Tolcher, A.W. et al. Phase I and pharmacokinetic study of CT-322 (BMS-844203), a targeted Adnectin inhibitor of VEGFR-2 based on a domain of human fibronectin. Clin Cancer Res 17, 363-71 (2011).

Discussion:

“We expect monobody adapters to minimally contribute to the overall immunogenicity of NPs, based on clinical trials with a related molecule (PEGylated Adnectin) (NCT02515669, NCT03984812^{44,45}).”

- Does the technology enable (more controlled and/or more facile) dual or triple targeting? Does that make it potentially standout among current NP surface modification technologies?

We thank the reviewer for this interesting suggestion. Theoretically, the Mb approach will allow dual or triple targeting. In the same way that this technique allows the use of off-the-shelf Abs, combining multiple Abs is greatly simplified compared to methods for which some prior chemical modification of Abs is necessary. We envision that the adaptor technology is particularly suited for optimizing the density and ratio of antibody combinations for effective dual targeting. We have added the following phrase to the discussion to incorporate this idea.

Discussion:

“Ab-Mb-NPs can also target different cell-surface molecules by simply swapping any available Ab of the same isotype or combining multiple of these Abs to hit different targets at once.”

Reviewer #2 (Remarks to the Author):

The authors present the use of a monobody as a chemical linker to immobilize antibody on the surface of nanoparticles. The Mb facilitates covalent attachment, proper orientation, and high loading density to the NP. The authors thoroughly characterize the conjugation and demonstrate the specificity of the Mb for mIgG-Fc binding. In addition, the authors compare this novel conjugation strategy to conventional EDC/NHS coupling chemistry. The results are quite impressive and address a critically needed method to improve conjugation of antibodies to NPs for highly functional bioconjugates that is also adaptable to many systems (unlike protein engineering). Most impressively, the authors show substantial improvement in conjugate binding in systems that are more appropriate models of clinical samples compared to the EDC/NHS prepared conjugates. Overall, the work is well-presented, highly significant, and scientifically sound. Several critiques are provided below, and minor revisions are recommended prior to publication.

1. The authors note that EDC/NHS prepared conjugates work in simple model systems (although less effective than Mb-mediated conjugation). However, the Mb conjugates provide significantly greater binding (>1000-fold) compared to EDC/NHC conjugates in more realistic setting. The authors attribute this improvement to better orientation and greater antibody loading when using Mb to prepare conjugates. It seems unlikely that these two factors alone can justify a 1000-fold difference in binding. If conventional conjugation provides 50% of the antibody loading and assume only ~30% are active/oriented, then the conventional conjugates still have 15% of the functional antibodies that the Mb conjugates present. Is it possible that the EDC/NHS linkage was not stable or effective (not covalent but electrostatic binding of antibody)? Alternatively, is it possible that multiple antibodies are required to bin the cells to tightly anchor the NP to the cell surface (multi-dentate binding), which would require a higher density ordered array of Ab on the NP surface? The data are compelling and sound, but it would be great to better explain the substantial improvement in NP binding for realistic samples.

We'd like to thank Reviewer #2 for their complimentary assessment of the rigor of our data. As mentioned in our response to Reviewer #1, the strength of the improvement in binding was surprising to us as well. Reports in the literature have investigated parameters that affect NP binding and shown a nonlinear (exponential) increase in binding as available targeting Ab increases above a certain threshold. This is likely due to eventually increasing multivalency of the NP, or creating more bonds simultaneously to ECs per NP as you suggest. This effect can show orders of magnitude increases in binding association constants (modeled³⁴, and reviewed³⁵). In addition, up to a 100-fold increase in antigen binding has been reported with an improved Ab orientation (along with evidence of poor orientation with EDC-NHS chemistry)¹⁹⁻²¹. In our work, with contributions from more properly oriented antibodies, as well as more total Abs (doubled), we hypothesize that the effective valency of the NP-cell interaction may increase, thus increasing the avidity of the NPs which explains the improvement in binding by multiple orders of magnitude. We have added text to the results section to better explain this observed increase in binding.

Results section:

“Together, our new Ab immobilization method dramatically increased targeted delivery of NPs compared with the conventional direct coupling in the multiple systems tested. We hypothesize that this improvement is due to substantial increases in the surface density of active Abs on the NPs (increased in number and by a more functional orientation), which has increased the instances of multivalent

interactions between cell-surface antigen and NPs. Increased multivalency is expected to increase binding efficiency exponentially^{34,35}.”

2. If the monobody is designed to universally bind mIgG1, will other mIgG1 molecules in a biological sample compete/displace the pre-adsorbed mIgG1 on the conjugate? This could be an issue when the Mb is redesigned to bind humanized antibodies and the conjugates are introduced into human samples which may contain many other hIgG proteins that could potentially result in antibody exchange and loss of function.

This is an important consideration. Our data in Figure S1 show that the dissociation of captured mIgG1 is very slow and thus its displacement is expected to be negligible. Moreover, since our primary application is *ex vivo* in perfusates that otherwise lack competing antibodies, this potential limitation is not likely to be an issue. Nevertheless, for future applications in humans, we envision that the use of a monobody adaptor specific to a particular mutant Fc, e.g. the so-called LALA variant, a format that is widely used for therapeutic antibodies, can eliminate this potential problem of IgG displacement. These points have been added to the text.

Discussion:

“We envision that the potential problem of Ab displacement with endogenous Abs can be eliminated with a monobody adaptor specific to a mutant Fc of human IgG (e.g. the so-called LALA variant⁴³ that is commonly used for therapeutic antibodies).”

3. The Ab and Mb loading on the NPs was measured using SDS-PAGE. More conventionally, total protein assays like BCA are used to quantify excess antibody in the supernatant. Can the authors comment/justify the choice of SDS-PAGE for quantitation and/or show calibration in supporting information? The precision and analytical sensitivity of this method would be valuable to show to the readers.

SDS-PAGE was selected to measure the Mb and Ab remaining in the supernatants for two reasons:

1. The SDS-PAGE allows us to differentiate the Mb and Ab on the basis of gel migration so we can determine if Mb are detaching from the NPs during the Ab association step; this would not be possible with a BCA assay that would not distinguish between Ab and Mb. (Results show no detectable amount of Mb detached from the Mb-NPs during the Ab association.)
2. There is a very small quantity of NPs remaining in the supernatant after centrifugation. These NPs are dye loaded and could affect the result of a colorimetric assay such as BCA. We have verified that residual NPs don't affect the SDS-PAGE result even if present in a concentration significantly higher than what remains in the tested sample. We have also tried to quantify the remaining Ab and Mb with a spectrophotometric assay (nanodrop) but the remaining fluorescent nanoparticles interfere with the measurement.

In addition to these points, the precision and sensitivity of SDS-PAGE is sufficient for the level of detection required here. We have added Supplementary Figure 2 with pictures of the gels, standard curve and sample measurement.

4. Can the authors comment on how the Mb compares to protein A/G-mediated conjugation that has previously been used to bind IgG-Fc and facilitate controlled orientation of antibodies bound to NPs?

While it is true that protein A/G-mediated conjugation facilitate the controlled orientation of the Abs, the attachment of the proteins A/G to the NPs is often not orientation controlled [1]. Many people use conjugation chemistries such as EDC-NHS to attach proteins A/G to NPs, which randomly orients the adapter. This shifts the orientation problem from the Ab to the binding protein and leaves part of the proteins unable to associate with Ab^{22,23}. On the other hand, our approach controls the Mb orientation due to the single cysteine specifically introduced to attach it to the NP. Thus, every Mb has the potential to associate with Ab.

Additionally, proteins A/G bind indifferently to various IgG subtypes (in particular protein A binds strongly to human IgG1, 2 and 4) as well as to serum albumin which would be problematic for the risk of displacement of the Ab after administration [2,3]. The Mb has limited dissociation and can be further engineered to bind to a non-naturally occurring Ab such as a mutant Fc as explained above in our response to question 2. Finally, proteins A/G can be highly immunogenic as they are derived from pathogenic bacteria, which likely limits their in vivo applications². As noted in our response to reviewer 1, this is less likely to be an issue with monobodies that are based on human protein sequences. We have added text to the introduction to address these points.

1. Iijima, M. & Kuroda, S. Scaffolds for oriented and close-packed immobilization of immunoglobulins. *Biosens Bioelectron* 89, 810-821 (2017).

2. Choe, W., Durgannavar, T.A. & Chung, S.J. Fc-Binding Ligands of Immunoglobulin G: An Overview of High Affinity Proteins and Peptides. *Materials (Basel)* 9 (2016).

3. Linhult, M., Binz, H.K., Uhlen, M. & Hober, S. Mutational analysis of the interaction between albumin-binding domain from streptococcal protein G and human serum albumin. *Protein Sci* 11, 206-13 (2002).

Introduction:

“However, well described limitations of this method include random orientation of the coupled Abs and potential modification of the antigen binding site. Both of these limitations lead to diminished binding potency due to a fraction of the Ab binding sites being unavailable¹⁸⁻²¹. **To partially mitigate this problem some have used Fc-binding proteins – such as protein A/G or FcRn to link the Ab and the NPs. These approaches only allow a partial control of the orientation as the protein themselves are randomly oriented at the surface of the NPs via EDC-NHS chemistry²²⁻²⁴.** As an alternative, targeting molecules can be engineered to support site-specific conjugation in a manner that avoids obscuring the sites of antigen binding (click chemistry, biotin-binding proteins, enzyme-based bioconjugation, ...).”

5. In Figure S1, mIgG1-Fc bound better than whole mIgG1. However, what if two full length mIgG1 molecules competed for FCM101 binding? (similar to comment 2)

We thank the reviewer for the question, but we'd like to clarify that Figure S1 does not include a comparison of the binding of mIgG1-Fc to whole mIgG1. Rather whole mIgG1 was added as a “chase” competitor of mIgG1-Fc, and the results shows little displacement in the presence of this competition. We have modified Panel B in Figure S1 to make this point clearer.

Furthermore, the values in graphs A and B cannot be compared as they were acquired using different assays. This point has been clarified in the caption.

Regarding competition of two full-length IgG molecules, we anticipate no difference between mIgG1 versus the Fc portion as tested, because the Fc portion is an independent module within IgG.

6. Figure S1 caption (b) “Yeast cells displaying FCM101 were first mixed with FCM101 (50 nM),” Should this be mixed with mIgG1-Fc (50 nM)?

We thank the reviewer for identifying this error, which we have now corrected it.

7. There are 9 Mb for every 1 Ab on the NP surface. Why not 1:1? Is this a size argument (antibodies are bigger and take up more surface area) or are some Mbs inactive, improperly oriented, etc.? Based on the surface area of NP and size of an Ab and Mb, what is theoretical Ab and Mb loading?

A 1:1 ratio of Ab to Mb is likely not ideal. Each Ab can be bound by 2 Mb due to the homodimeric architecture of the Fc region, and this multivalent interaction (avidity) results in strong linkage with slow dissociation, so at a minimum, a 2:1 ratio (Mb:Ab) is anticipated. Moreover, because IgG (~150 kDa) is a much larger molecule than Mb (~10 kDa excluding flexible tails), it is plausible that there is not enough space on the NP surface to allow a 1:1 Mb to Ab ratio. We also have no reason to think that the Mb could be improperly oriented on the surface of the NPs. Furthermore, additional available Mb (beyond a 2:1 ratio) can allow for rebinding of dissociated Fc, which further reduces the dissociation of Ab from the NP surface. It is unclear what the highest ratio can be before we lose bivalent binding and sufficient rebinding.

As to the theoretical maximum of Mbs attaching to the NP surface, the limiting factor is the amount of maleimide available and not the surface area. The total number of maleimide groups is around 9380 maleimide/NP if all maleimide groups are located on the surface of the NPs. We currently reach 1120 Mb/NP, but this corresponds to a conjugation efficiency of 45% of added Mb, so other unidentified parameters must be involved. These calculations are now detailed in Supplementary Figure 4.

8. Figure 2 heading for A-D “Cell culture under flow” should be “Static Cell Culture”.

Yes, this was our error, which we have now corrected.

9. Caption for Figure 3 (F-I) reads as if all samples are PAECs that are incubated with different labels; however, Figure and text suggests samples differ (PAECs and HUVECs) while labels are the same.

We thank the reviewer for noticing. The caption was correct and we have now changed the figure to reflect what each image represents.

Reviewer #3 (Remarks to the Author):

Overall the paper is concerned with optimising a previous study where mAbs were conjugated to nanoparticles. Monobodies are introduced without the context of previous work such notably the adnectin-anti-VEGFR2 paper (first use of introducing a cysteine for PEG conjugation), or the centyrin work (inserted a cysteine at every position to find the best site for conjugation).

We agree with the reviewer that it is beneficial to include additional context for introducing the monobody technology. As this reviewer points out, there is unfortunately substantial confusion about the history of monobodies and related systems. Monobodies, first reported in 1998, are the founding system of the many synthetic binding proteins based on a fibronectin type III domain²⁶⁻²⁸. This confusion is due partly to

the fact that subsequent systems were introduced without sufficient context, including adnectins, which licensed the monobody technology from the Koide group [1] and centyrins [2]. We have added additional introduction to the Mb technology and its relationship with the subsequent related technologies.

Regarding the introduction of a cysteine to the monobody/adnectin construct, we have added a sentence describing the precedents (the adnectin-anti-VEGFR2 paper for PEG attachment³², and Wojcik et al by the Koide group for fluorescent dye attachment³⁰, both in 2010). We also refer to the centyrin work³¹ (2016), as recommended, although it came much later than these 2010 studies but is relevant to this work.

1. Xu, L. et al. Directed evolution of high-affinity antibody mimics using mRNA display. Chem Biol 9, 933-42 (2002).

2. Jacobs, S.A. et al. Design of novel FN3 domains with high stability by a consensus sequence approach. Protein Eng Des Sel 25, 107-17 (2012).

Results:

“Mbs are the founding system of the many synthetic binding proteins based on a human fibronectin type III domain²⁶⁻²⁸. Mbs are well described and are developed from combinatorial phage-display libraries in which select residues are diversified using highly tailored amino acid compositions, followed by gene shuffling and further screening in the yeast-display format²⁹.”

“We next exploited the fact that Mbs are inherently free of cysteines and introduced a single cysteine residue in the extended C-terminal tail for site-specific conjugation (Fig 1a) as has been shown in our previous work and by others³⁰⁻³².”

The authors have created a nanoparticle approach that ensures conjugated antibodies are outward-facing. They demonstrate:

- Using monobodies as an adapter to connect anti-ICAM2 mAbs with PLA-PEG nanoparticles, which increases nanoparticle binding to target epithelial cells over NPs with the mAb directly conjugated
- Then swapping the anti-ICAM2 mAb for an anti-CD31 mAb and demonstrating specific binding to the new target
- Then swapping the PLA-PEG to a PACE-PVA nanoparticle
- Then flushing a human kidney (rejected for transplant) with anti-CD31-Mb-NPs and demonstrating specific binding

We would like to respectfully clarify some points made in this above summary.

- We primarily use anti-CD31 mAbs to target endothelial cells and not anti-ICAM2 mAbs to target epithelial cells.
- The transplant-declined human kidneys were not flushed with anti-CD31-Mb-NPs. Instead the kidneys were perfused using normothermic perfusion machine with a washed red blood cell based perfusate that matches what is currently in clinical use in the U.K. The CD31-Mb-NPs were injected into the perfusate and perfusion was continued for 4h.
- We have used organs which have been declined for transplant and not rejected as “rejection” has a different meaning in the transplantation literature. This distinction is important as the therapy we hope to develop would reduce post-transplant rejection of suboptimal organs that are currently declined for transplant because of the high likelihood of post-transplant rejection.

Generally, I find this technology to be overly complicated:

1. I don't understand the significance of monobodies in this technology:

- The full antibodies could be conjugated via cysteine chemistry (e.g. <https://www.sciencedirect.com/science/article/abs/pii/S0168365920300547> from 2020),
- or cut down to the ScFvs and conjugated via cysteines (e.g. <https://pubs.acs.org/doi/abs/10.1021/ja0555668> from 2006)
- The monobody could be alternatively replaced with a protein G, FcRn or other Fc-carrying protein

The main advantage of the Mb over the above suggestion is that no modification of the Ab is needed to control the orientation on the NPs. The Mb is engineered to present a single cysteine residue for site-specific conjugation to the NP surface. Once we have optimized the Mb conjugation reaction, no additional chemical steps need to be optimized for different antibodies with the same isotype or for different batches of NPs. There are several additional points of comparison:

- **Binding through native cysteine:** As the reviewer points out, the full Ab could be conjugated via cysteine chemistry. This method mainly targets cysteine residues that form the disulfide bonds in the hinge region and/or the C-terminus of the Fab region. These disulfide bonds can be reduced and then used to conjugate the full antibody in a region-specific manner. Although this strategy allows for orientation control, the disulfide bonds need to be carefully reduced to produce the specific cysteine residue at the desired location. The sensitivity of these cysteines to a reducing reagent can vary for different antibodies, so each Ab needs to be studied independently. Equally important, the product is heterogeneous and it is not trivial to achieve high reproducibility of chemical conjugation. Furthermore, there is often a substantial loss of materials in the preparation.
-
- **Conjugation of scFv via cysteines:** it is technically feasible to produce scFvs and conjugate them via cysteines, although this method suffers from the same limitations as the conjugation of the full antibodies via cysteines, as outlined above. Moreover, very few scFvs are commercially available, whereas the library of available intact antibodies is enormous. Therefore, in most cases, one would have to develop their own scFv reagents.
-
- **Binding through other Fc-binding proteins such as protein A/G or FcRn:** replacing the Mb with the other molecules suggested could also control the orientation of Ab, but the attachment of the proteins A/G or FcRn to the NPs is not orientation controlled. Many laboratories use chemistry such as EDC-NHS, to attach the adapter protein which randomly associates a part of the protein to the surface of NPs. This shifts the orientation problem from the Ab to the binding protein and leaves part of the proteins unable to associate with Ab^{23,24}. Furthermore, proteins A/G are less specific than monobodies and bind various IgG subtypes (in particular protein A binds strongly to human IgG1, 2 and 4) as well as serum albumin that may be problematic for the risk of displacement of the Ab after administration, particularly in applications that require the use of serum or blood [1,2]. On another hand, FcRn is a large protein with lower affinity to the Fc region than FcRn1. FcRn1 has limited dissociation and could be further engineered to bind to a non-naturally occurring Ab such as a mutant Fc.
-
- Furthermore, as superantigens from pathogens, protein A/G binds to many Fab frameworks in addition to the Fc region, which can prevent consistently oriented presentation of IgGs.

We have added text to the introduction that emphasizes the improvement of the monobody approach compared to existing surface modification methods.

1. Choe, W., Durgannavar, T.A. & Chung, S.J. Fc-Binding Ligands of Immunoglobulin G: An Overview of High Affinity Proteins and Peptides. *Materials (Basel)* 9 (2016).
2. Linhult, M., Binz, H.K., Uhlen, M. & Hober, S. Mutational analysis of the interaction between albumin-binding domain from streptococcal protein G and human serum albumin. *Protein Sci* 11, 206-13 (2002).

Introduction:

“However, well described limitations of this method include random orientation of the coupled Abs and potential modification of the **lysine residues crucial for antigen binding**. Both of these limitations lead to diminished binding potency due to a fraction of the Ab binding sites being unavailable^{18,21}. **To partially mitigate this problem some have used Fc-binding proteins – such as protein A/G or FcRn to link the Ab and the NPs. These approaches only allow a partial control of the orientation as they are themselves randomly oriented at the surface of the NPs via EDC-NHS chemistry²²⁻²⁴. This might negatively impact the surface density of the targeting Ab if not all adapters are available for binding.**

As an alternative, targeting molecules can be engineered to support site-specific conjugation in a manner that avoids obscuring the sites of antigen binding (**click chemistry, biotin-binding proteins, enzyme-based bioconjugation, and others**). **These alternative strategies are comprehensively reviewed by Sivaram et al¹⁸. While effective, these require modification of each Ab which is labor intensive and costly, diminishing the ability to rapidly screen antibodies targeting different antigens. During the early development phase of a new targeted NP it is essential to identify the optimal antigen and targeting reagent for the given species, disease, organ and cell. A method to control Ab orientation without having to re-engineer each individual targeting molecule under consideration would be impactful.** Here we describe a new approach that combines simplicity, adaptability and efficiency in binding **using unmodified antibodies as the targeting molecule.**”

2. In the case that the monobody is somehow essential, why attach an antibody?

In theory, a monobody could be designed that binds a specific cell surface ligand. The strength of using a Mb as an adapter to bind and attach Ab and the NPs is that it provides the plug-and-play modularity that allows access to the many different Ab available. We describe our system as a building block design composed of three distinct blocks: 1) the drug-nanoparticle block that can be changed according the therapeutic need, 2) the targeting Ab block that can be changed as the target changes between diseases, and from cells to animals to humans and 3) the Mb block which stay constant to universally link block 1 and 2 together. This design greatly simplifies the development phase by allowing the use of unmodified Abs. We suggest the experiment shown in Figure 3, where the Ab is easily switched between off the shelf Abs without any further optimization or modification demonstrates this added value. As R&D and preclinical research progress, new targets can be easily assessed without investing time and money into special design and production of Abs or derivatives as shown in some extent in Figure 3. In addition, once a targeting strategy is in place the drug-nanoparticle can be swapped for alternate combination to treat a different pathology in the same cell type. We demonstrate this in Figure 4 where the same targeting strategy (i.e. same Ab and Mb) is successfully applied to PACE polymer which is optimized for nucleic acid delivery.

Introduction:

“We have also demonstrated the versatility of this approach by showing how the Mb can enable easy exchange of both the NP and targeting Ab without sacrificing binding efficiency. **In other words, Ab-Mb-NPs are composed of three distinct blocks: 1) the drug encapsulating nanoparticle, 2) the Mb adaptor and 3) the targeting Ab. The Mb block serves as a universal adaptor that links diverse combinations of NP and targeting Ab blocks that can be chosen to fit both the therapeutic need and corresponding target.**”

Discussion:

“Ab-Mb-NPs can also target different molecules or species by simply swapping any available Ab of the same isotype. **This demonstrates an improvement over the strategies which require Ab modification.** The exchange of the targeting Ab could be motivated by either a transition between pre-clinical models (e.g. porcine to human), or could be used to facilitate the screening and discovery of the optimal antigen to target for a particular tissue. In addition to the versatility of the targeting molecule, the Mb can be attached to a variety of NPs presenting a thiol reactive group on their surface. This approach grants the ability to choose the best vehicle for any selected therapeutic without further modifications to the targeting strategy and expands the possible applications. **In the end, this approach can be thought of as a building-block design where each critical component – therapeutic agent, nanoparticle and antibody - can be switched without modifying the other blocks. Thus, we envision that our approach will facilitate more comprehensive screening for the optimal combination of these three components.**”

- The antibody format used is from a mouse, which will cause further immunogenicity in a transplanted organ

We agree with the reviewer on this point. A Mb targeting mouse IgG1 Fc is used here as proof-of-concept because of the wide availability of research Abs in the mouse IgG1 format. A human or humanized Ab would be used for translational studies using a Mb developed for this purpose. As is described in the discussion:

“To minimize immune-reactions in humans, we envision the use of human or humanized Abs and a Mb adapter specific to human IgG Fc. Just as human Abs can be developed using many modern technologies, Mbs to human Fc can be developed following the well-established pipeline^{27,29}. Furthermore, we envision the potential problem of Ab displacement with endogenous Abs can be eliminated with a monobody adaptor specific to a mutant Fc of human IgG (e.g. the so-called LALA variant) that is commonly used for therapeutic antibodies⁴³.”

- The authors already argue that generating a new monobody with binding to different human IgG types is not a challenge, so why not just raise a monobody directly against your target of interest? o i.e. their words “Just as human Abs can be developed using many modern technologies, Mbs to human Fc can be readily developed following the well-established pipeline^{12, 21}.”

We thank the reviewer for the pointing out the need for a more nuanced description on the effort to generate new Abs and Mbs. A single adapter Mb that binds to an IgG subtype will allow us to examine the effectiveness of targeting different antigens using existing antibodies. Developing a single Mb to bind a particular Fc region (as we have done here) is much less work than developing multiple high-quality Mbs against the many possible antigen targets. We agree that “can be readily developed” is an overstatement. The reality is that, though there are established pipelines, it is still labor intensive (3-6 months of dedicated effort are needed for the development of a new Mb). Our approach circumvents such efforts and expenditure in identifying an effective antigen and/or effective NP for a particular indication. Once an

optimal design (NPs-drug + target) is found, the development of a Mb that directly binds to the antigen of interest is well justified. We have revised the text to further clarify the advantage of the adaptor system in the preclinical studies.

- Of note, inserting a cysteine into monobodies for conjugation is also not a novel approach (e.g. <https://pubmed.ncbi.nlm.nih.gov/27737926/> or <https://www.ncbi.nlm.nih.gov/pmc/articles/PMC2840239/>) which have not been mentioned despite being core works in the field.

We did not intend to claim the novelty of this approach, particularly as we ourselves had done it in 2010. Please see our response above. We have modified the text in Results to acknowledge this point.

Results:

“We next exploited the fact that Mbs are inherently free of cysteines and introduced a single cysteine residue in the extended C-terminal tail for site-specific conjugation (Fig 1a) **as has been shown in our previous work and by others³⁰⁻³².**”

Overall, interesting study but is effectively just an optimisation of their previous work (<https://stm.sciencemag.org/content/9/418/eaam6764.short>), and uses a technology that adds the complexity of an adapter, which could be avoided by instead optimising the initial IgG conjugation chemistry. Overall, I think it’s an interesting, detailed and carefully performed study but maybe not groundbreaking.

We wish to emphasize that the approach described in this paper is not an iteration of our previous work but a new technology that could be used to efficiently target nanoparticles. We respectfully disagree that this could be achieved by optimization of IgG chemistry, which—as we pointed out—eliminates modularity, the control of the orientation and/or high reproducibility. In our experience, the use of a Mb does not add complexity, but reduces it by introducing modularity to simplify pre-clinical development. We have modified the text to emphasize and illustrate these points.

REVIEWERS' COMMENTS

Reviewer #1 (Remarks to the Author):

The authors have properly addressed my comments and questions. The added experiments and textual clarifications have (further) improved the manuscript.

Reviewer #2 (Remarks to the Author):

The authors have satisfactorily addressed concerns raised in the initial review, and the revised manuscript is recommended for publication.